# Hybrid AI-enhanced lightning flash prediction in the medium-range forecast horizon

Mattia Cavaiola[1,2,4] ✉, Federico Cassola[3], Davide Sacchetti[3], Francesco Ferrari[1,2] & Andrea Mazzino [1,2] ✉

Traditional fully-deterministic algorithms, which rely on physical equations and mathematical models, are the backbone of many scientific disciplines for decades. These algorithms are based on well-established principles and laws of physics, enabling a systematic and predictable approach to problem-solving. On the other hand, AI-based strategies emerge as a powerful tool for handling vast amounts of data and extracting patterns and relationships that might be challenging to identify through traditional algorithms. Here, we bridge these two realms by using AI to find an optimal mapping of meteorological features predicted two days ahead by the state-of-the-art numerical weather prediction model by the European Centre for Medium-range Weather Forecasts (ECMWF) into lightning flash occurrence. The prediction capability of the resulting AI-enhanced algorithm turns out to be significantly higher than that of the fully-deterministic algorithm employed in the ECMWF model. A remarkable Recall peak of about 95% within the 0-24 h forecast interval is obtained. This performance surpasses the 85% achieved by the ECMWF model at the same Precision of the AI algorithm.

Lightning flashes are severe threats to wildfires, aviation, tele-communication, electrical infrastructures, and, more generally, to human life[1]. Studies conducted on the assessment of the worldwide impact of lightning concluded that 24,000 deaths and 240,000 injuries occur per year[2]. Moreover, due to global warming, evidence that the occurrence of lightning has increased has been reported[3]. Accurate forecasts of these extreme events are thus of crucial importance for decision-makers.

Nowadays, lightning forecasts are based on parameterization schemes encoded in the numerical weather prediction (NWP) models[4,5], i.e. deterministic prediction models based on partial differential equations. NWP models excel in providing accurate forecasts for medium to long-range time scales, as they capture the underlying physical laws and interactions within the atmosphere[6]. They are especially reliable for predicting large-scale weather patterns and phenomena. However, solving complex differential equations requires

significant computational resources[7]. Over the years, several semi-empirical parameterizations of lightning flashes have been developed for numerical cloud models[8]. Nevertheless, there is still disagreement about molecular and microphysical processes responsible for initiating a lightning flash, including the role of turbulence and in particular of large velocity excursions[9]. A similar understanding of the role of turbulence obtained in cloud microphysics[10] is thus still far from being achieved in the realm of lightning theory. Theories and laboratory experiments developed in recent years have formed the basis of modern lightning parameterizations[8,11]. However, to set all free parameters over a region of interest, a large amount of observed data is needed, a requirement often difficult to satisfy, lightning flashes being rare events. In mid-2018, the European Centre for Medium-range Weather Forecasts (ECMWF) introduced lightning flash density forecast on a global scale in its operational HRES NWP model[12,13]. In ref. 14 authors evaluate the ECMWF-HRES lightning forecasts over India

[1]DICCA, Department of Civil, Chemical and Environmental Engineering, Via Montallegro 1, Genova 16145, Italy. [2]INFN, Istituto Nazionale di Fisica Nucleare, Sezione di Genova, Via Dodecaneso 33, Genova 16146, Italy. [3]ARPAL, Regional Agency for Environmental Protection Liguria, Genova, Italy. [4]Present address: CNR - National Research Council of Italy, Institute of Marine Sciences, Via S.Teresa S/N, 19032 Pozzuolo di Lerici, La Spezia, Italy. ✉e-mail: mattia.cavaiola@sp.ismar.cnr.it; andrea.mazzino@unige.it

during the pre-monsoon season of 2020 using lightning observation data. However, this analysis was limited to a period of only 3 months during which only intense and frequent lightning activity occurred.

Alternative strategies to deal with the issue of forecasting exploit statistical and artificial intelligence (AI)-based predictions. We use the definition of AI-based methods to specify data-driven forecasting systems, i.e. forecasts made solely in terms of the knowledge of field data. The task of generating the ruling dynamics is done in this case by the AI which learns how to map observations (or mix of observations and model-reconstructed fields) into target meteorological fields at different lead times. Along this line, recently, AI-based methods[15] have shown potential in accelerating weather forecasting by orders of magnitude, even if the accuracy of the forecasts is not always higher than that of NWP methods. Very recently, however, an AI-based method, dubbed Pangu-Weather, was introduced in ref. 16, the accuracy of which has been compared with the world's best ECMWF-HRES NWP model obtaining stronger deterministic forecast results. In that strategy, revolutionary in a sense, AI learns the weather dynamics from ERA5 global reanalysis[17]. Target variables are thus restricted to those considered in the ERA5 database (or to a smaller subset as done in ref. 16). In way of example, lightning flashes are not included in the forecast chain. Also, ERA5 can hardly be interpreted as the ground truth of the atmosphere. Indeed, ERA5 combines observations and model simulations to create a consistent and continuous dataset of atmospheric variables. In this respect, errors are expected (and actually identified in ref. 18 where energy-related applications in regions of complex orography and/or complex sea–land interactions have been studied using ERA5) in reconstructing fine-structure weather features. Because the network presented in ref. 16, at least in its present form, only learns weather features from ERA5, the same skills shown in ref. 16 must not be taken for granted when the focus is small-scale weather features or extreme events[19]. We will discuss further (see the section "Discussion") how Pangu-Weather could be used in synergy with our approach to achieving lightning occurrence predictions at an unprecedented level of accuracy.

AI-based strategies have been widely used in recent years to predict lightning occurrences only a few hours ahead (nowcasting). Among the numerous works in the literature that tackle the problem of lightning flash forecast from the nowcasting perspective, we mention some relevant recent examples[20–24]. Very recently, in ref. 20, authors present NowcastNet, a neural-network framework with end-to-end forecast error optimization, which provides skillful forecasts at light-to-heavy rain rates, associated with convective processes that were previously considered intractable. In ref. 24, authors address the nowcasting of extreme weather events by fully data-driven AI algorithms. There, an ensemble method, based on binary classification of extreme events characterized by a high level of precipitation and lightning density, has been applied. The forecast of thunderstorms is addressed in ref. 25 via a binary classification that exploits a convolutional neural network based on satellite images and lightning recorded in the past. They achieve a probability of detection of more than 94% for 15-min ahead lightning forecasts. Another example of binary lightning nowcasting is given in ref. 26, where authors use a deep learning framework in order to forecast lightning flashes one minute ahead. In ref. 27, the authors propose a semantic segmentation deep learning network for cloud-to-ground (CG) lightning nowcasting, named LightningNet. This network is based on multisource observation data, including data from a geostationary meteorological satellite, Doppler weather radar network, and CG lightning location system. Results show the ability of LightningNet to achieve good performances in forecasting lightning flashes in the 0–1 h horizon using the multisource data.

In ref. 28 authors integrate observation and data coming from the NWP model in a method based on a dual-encoder. The first step of this method consists of extracting information about the spatio-temporal

distribution of lightning. To do that, both simulations from the Weather Research and Forecasting (WRF) model[29] and observations have been considered and successively inputted to a dual encoder. The extracted features are then merged and serve as input to a spatio-temporal decoder to make predictions via convolutional long short-term memory (LSTM) up to 6 h ahead. Finally, in[30], authors propose an attention-based dual-source spatio-temporal neural network (ADSNet) for 12-h ahead lightning forecast. A data-driven neural network is used for hourly lightning forecasts, which exploit both the WRF simulations and the recent historical lightning observations.

The vast majority of existing AI-based lightning forecast strategies are thus fully data-driven and the focus is restricted solely to nowcasting. This is a serious limitation in view of the many vital applications related to forecasting lightning flashes some days in advance.

There is thus a research gap within the current body of literature. To contribute to filling this gap, we present a deep learning framework, here referred to as FlashNet, able to forecast the lightning flash occurrence up to 48 h ahead in terms of binary classification. FlashNet uses features coming from the HRES NWP global model of the integrated forecasting system (IFS) of ECMWF. The results are validated against the Italian lightning observation network (LAMPINET)[31,32], detailed in the "Methods" section, and we use the lightning flash predictions provided by HRES (see the "Methods" section) as a relevant benchmark of our forecast. Our results show a clear added value of our AI-enhanced, hybrid, strategy, over the fully deterministic approach exploited in the HRES model for lightning forecasting.

To summarize the current situation, two distinct paradigms of weather forecasting are available, each with its own level of maturity, strengths, and weaknesses. As it emerges from ref. 33, the present situation is more like a competition between AI-based methods and classical NWP models, rather than a synergistic approach. Here, we will not contribute to finding a winner between AI and NWP models. Rather, we propose a way to integrate AI and NWP models based on the belief that their integration holds the potential to revolutionize weather forecasting by combining physics-based understanding with data-driven insights, ultimately improving our ability to predict and understand the Earth's complex atmospheric behavior.

According to the discussions alluded to above, we have classified our strategy as an AI-enhanced strategy where the AI acts as a post-processing method downstream from the predicted future state of the atmosphere. Learning the mapping between large-scale features predicted by HRES and the ground truth of lightning flashes we are able in this way to predict lightning flash occurrences where observations explicitly enter into play bringing a significant added value.

A further advantage of our strategy is that it remains free of the three potential risks identified in ref. 19 related to AI exploitation for weather predictions. Namely, (i) undersampling of 'monster storms' occurring only a few times a century; (ii) problems in predicting meteorological features, such as severe storms, fronts, or tropical cyclones; (iii) possible occurrence of highly erratic predictions when the AI strategy encounters conditions never encountered before. The reason is that in our strategy AI is exploited downstream of forecasts from NWP models: these latter are well-known to be skillful on all three points raised in ref. 19.

## Results

Before delving into the discussion of the results, it is useful to provide an overview of the methodology that led to these outcomes. Detailed information can be found in the "Methods" section.

Our approach is based on two fundamental pillars: the availability of spatially distributed and accurate lightning observations and the availability of proxies extracted from the output of the global NWP model ECMWF-HRES, considered important for lightning initiation. These two pillars are connected through an AI network capable of learning the highly nonlinear mapping between HRES features and

observations. Instead of assessing the robustness of the learning process by dividing the entire period in which observations and proxies are available for this study (from 2019 to 2021) into training, validation, and test sets, we opted for a validation strategy that allows for controlling the robustness of the inference phase, thus avoiding conclusions that are too dependent on the specific choice of the test year. The tool we deemed most suitable for this purpose is the *k*-fold cross-validation (here $k = 3$), where possible biases caused by a fortunate/unfortunate choice of the test year are mitigated by selecting all possible years in turn. For each fold of the *k*-fold cross-validation, the predictive skills of our AI strategy will be measured in terms of the classifier's ability to correctly classify the presence and absence of lightning events. The classifier's ability will be rigorously analyzed, considering fundamental properties such as its reliability, sharpness, and, more generally, its capacity to minimize false positives and false negatives.

The forecast skills will be evaluated for the entire area of interest depicted in Fig. 1, corresponding to 9975 points covering an area of 997,500 km² where our predictions will be compared against both the ground truth and HRES' predictions.

Two datasets will be analyzed in the paper: one representing the original level of imbalance, characterized by a lightning occurrence fraction of ~1%, and another that has been artificially balanced to ensure an equal distribution of lightning presence and absence. The balanced dataset enhances the learning process allowing the network to gain insights into both event and non-event scenarios with equal representation. The results we are going to discuss, concerning the assessment of the model's skills, will pertain to both datasets, providing a comprehensive evaluation of our approach under varying conditions.

**Measuring FlashNet's success of prediction**

Figure 2a reports the Precision–Recall curves (see the "Methods" section) obtained from FlashNet's predictions on both the balanced and the unbalanced test sets (see the "Methods" section). These curves have been produced by accumulating all 3-hourly forecasts in the 0–24 and 24–48 h forecast intervals. The dependence of Precision score on Prevalence[34] (see the "Methods" section for the explicit formula relating the Precision scores of datasets having different Prevalence) is clearly detectable from the figure. The same dependence also reflected in the Area-Under-the-Curve (AUC) score, corresponding to the integral of the Precision–Recall (P–R) curve (see the "Methods" section), reported as a legend of Fig. 2a, both for the balanced and the unbalanced dataset. AUC score dramatically reduces passing from the balanced test set (AUC = 0.93, for the test in 2021 in the 0–24 h forecast interval) to the unbalanced one (AUC = 0.18 for the test in 2021 in the 0–24 h forecast interval). Contrary to what may be inferred from Fig. 2a, it is not at all true that our model's performance is strongly dependent on the dataset's imbalance. In fact, the opposite is true: once trained in a balanced training set, our predictive model performs comparably whether it is applied to a balanced or an imbalanced test set. The crux of the matter is discussed in the "Methods" section (Eq. (8)). This formula, which does not appear to be documented in the literature, explains how the P-R curve scales when transitioning from one prevalence value (and therefore, a specific level of imbalance) to another while keeping the model's predictive abilities invariant by changing the prevalence. Unlike precision, note that the Recall value is invariant by changing the dataset level of imbalance (again assuming model-invariant skills by changing prevalence). The low AUC values in the case of the imbalanced dataset are simply a consequence of the high dataset imbalance, not an indication of the poor predictive

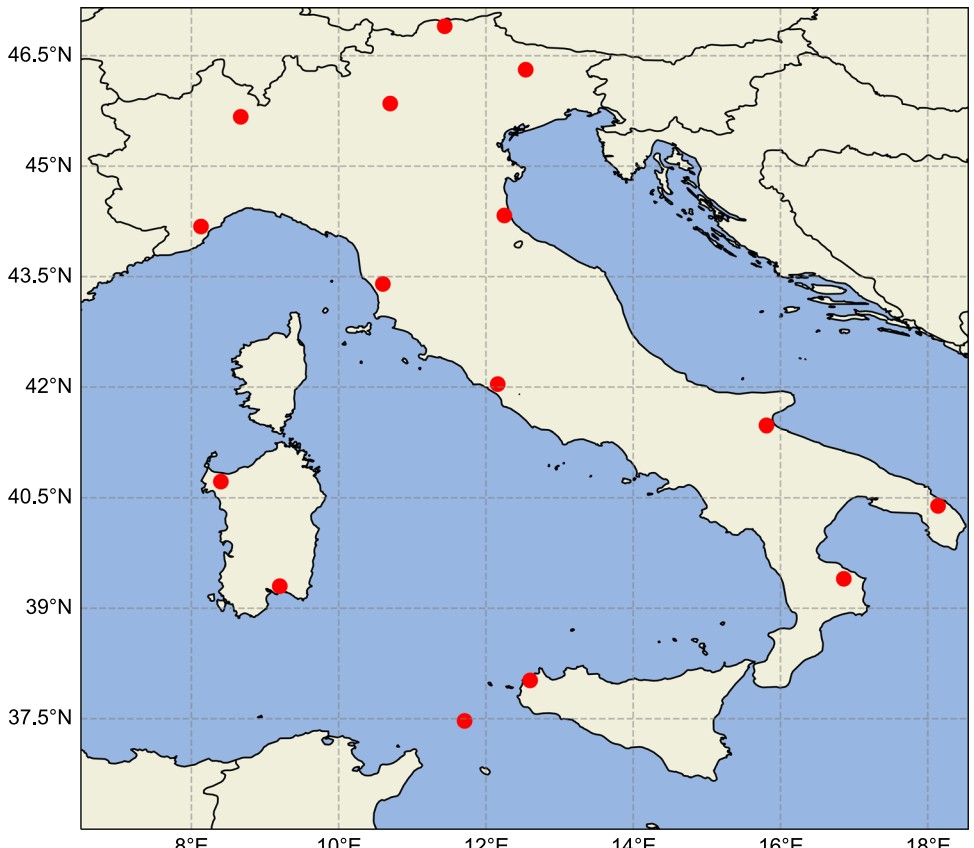

**Fig. 1 | The LAMPINET detection system.** The case study area is reported together with the locations of IMPACT ESP sensors (red circles) used to detect lightning flashes in the LAMPINET detection network.

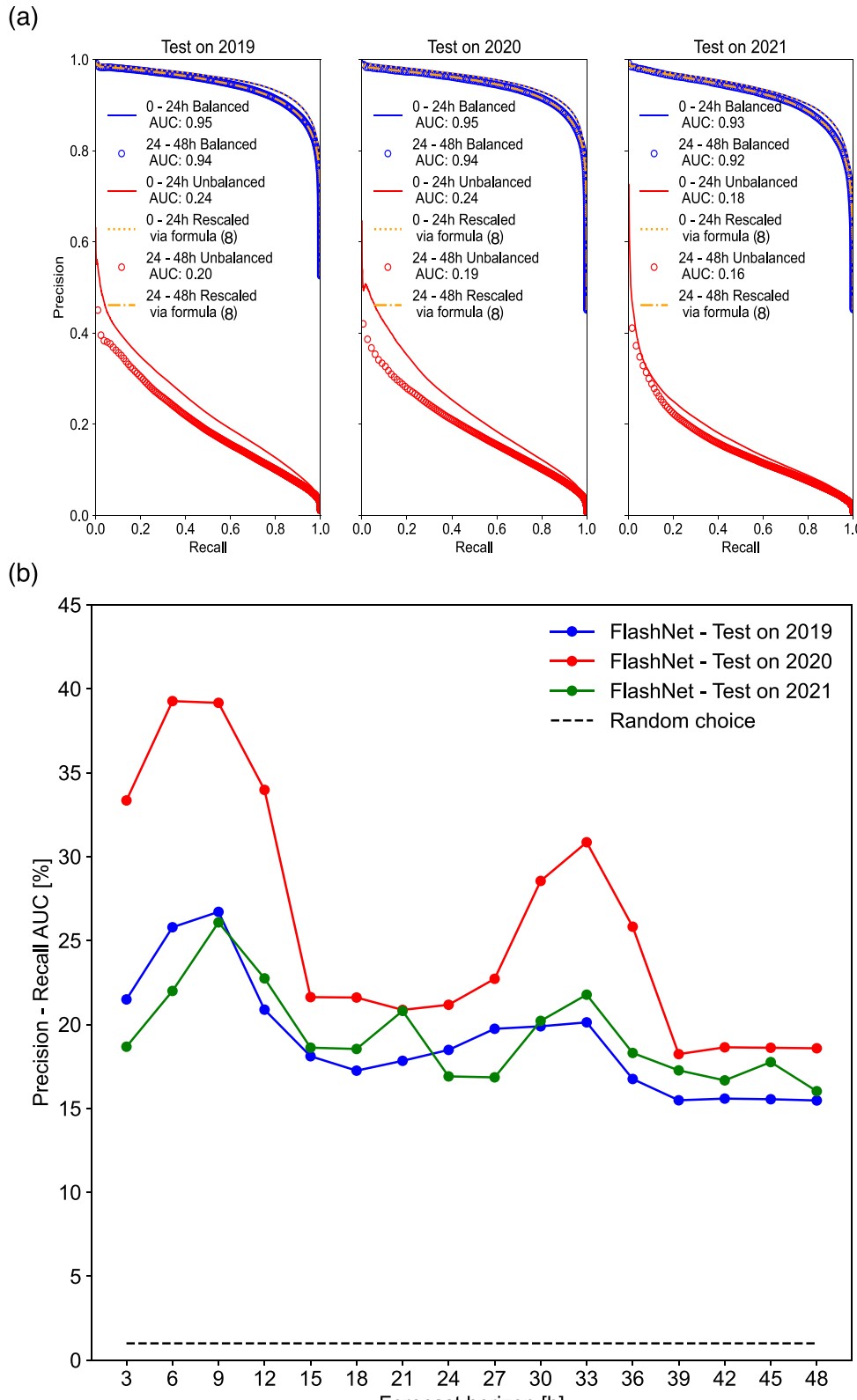

capabilities of our model. To demonstrate the validity of this statement, we use Eq. (8) to rescale the P–R curve from all cases with small prevalence to the corresponding balanced cases. The rescaled curve is depicted as solid orange lines (dashed and dot-dashed) and closely aligned with the corresponding blue lines/open circles. This serves as evidence for the truthfulness of our earlier assertion.

It is also worth interesting that passing from the 0–24 h to the 24–48 h forecast interval only slightly deteriorates the AUC score. This is an indication that the strategy might be successfully extended to longer forecast horizons.

Figure 2b shows the AUC for each forecast lead time, separately obtained from the three years of the 3-fold cross-validation (red, blue,

**Fig. 2 | Assessing FlashNet's predictive skills. a** *Precision−Recall* curves for the balanced test sets (blue curves/markers) and the original unbalanced test sets (red curves/markers). Continuous lines refer to the predictions in the 0–24 h forecast horizon; open circles refer to the 24–48 h forecast horizon. The three panels refer to the three folds of the 3-fold cross-validation considered in the present study. The unbalanced dataset Prevalence is: 1.2% (year 2019), 0.9% (year 2020), and 0.9% (year 2021). All balanced datasets have a Prevalence close to 50%. All curves corresponding to all imbalanced cases (red lines/symbols) have been rescaled according to Eq. (8) to the prevalence value of the corresponding balanced cases

(blue lines/symbols). The rescaled curves are in orange, dash-dotted/dotted lines. When doing that, an almost perfect superimposition between the curves is observed, confirming FlashNet's capability to work properly in both balanced and imbalanced cases. **b** The AUC score is reported normalized by the Prevalence as a function of different forecast lead times for the three folds of the cross-validation. The blue, red, and green lines are from FlashNet's forecasts; the dashed line corresponds to the forecasts obtained from the Prevalence-based random model discussed in the main text. Source data are provided as a Source Data file.

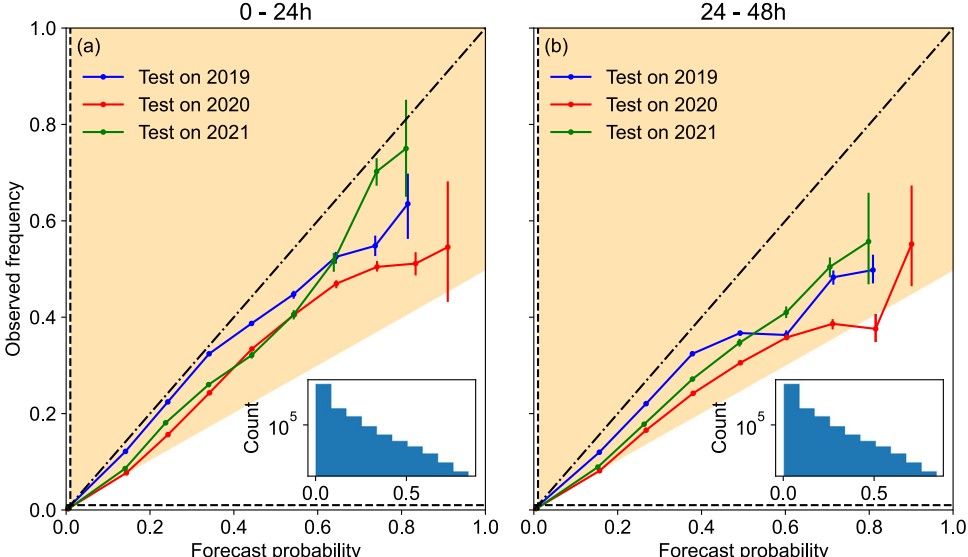

**Fig. 3 | Assessing FlashNet's reliability.** The reliability diagram is shown for the three unbalanced folds of the 3-fold cross-validation we have considered in the present study. Confidence bars are also shown and correspond to the 95% bootstrap confidence interval around the mean. Panel **a** covers the forecast horizon from 0 to 24 h, while panel **b** spans the 24–48-h period. Shaded regions show the area of skillful forecasts. The horizontal and vertical lines show the prevalence-

based probabilities of the event for forecasts and observations. The two insets show the sharpness diagram (relative to the test in 2021) illustrating the relative frequency of lightning occurrence predictions (abbreviated as 'Count') at different probability levels. These two diagrams also provide the number of sampling (Count) used to compute the 95% bootstrap confidence interval alluded to above. Source data are provided as a Source Data file.

and green lines). Here, the AUC is normalized with the Prevalence (see the "Methods" section) evaluated for each forecast lead time. From Fig. 2b, the added value brought out by FlashNet is largely appreciable, the resulting AUC reaching values up to about 40 times the one corresponding to the random model detailed in the "Methods" section (red line to be compared with the black dashed line). Also of interest from Fig. 2b is the correlation between the resulting AUC and the diurnal cycle, with lower skills occurring during nighttime hours compared to daytime hours. As we will explore further in the following sections, this characteristic appears to be a consequence of the diurnal cycle-dependent nature of HRES's skills.

### Assessing FlashNet's reliability

FlashNet's forecast reliability is assessed via the reliability diagram introduced in the "Methods" section. The plot is reported in Fig. 3 for the two forecast horizons 0–24 h (left panel) and 24–48 h (right panel). The bisector corresponds to perfectly calibrated forecasts (defined in the "Methods" section) while the shaded region represents the skill area[35] indicating that the forecasts below the dashed horizontal line are not better than the reference forecast based on Prevalence.

As one can easily see, our FlashNet network produces forecasts well within the skill area, the hallmark of reliability, even if a tendency to overestimate lightning occurrence can be detected: the actual proportion of lightning events is generally lower than estimated. Only a single point displays a mean reliability value just outside the skill area

(year 2020, forecast lead time 24–48 h). However, the 95% bootstrap-derived error bar does overlap with the skill region.

Overall, according to the categorization of reliability reported in Fig. 2 of ref. 35 the reliability of FlashNet can be defined as "very useful for decision making".

Also shown in the two panels of Fig. 3 (as insets) is the sharpness diagram relative, for the sake of example, to the year 2021. Sharpness diagrams[36] show the relative frequency with which the lightning occurrence has been predicted (over the test set) with different levels of probability. In both insets, the majority of forecasts predict low probabilities for lightning occurrence. Our network is also capable of predicting relatively high probabilities of lightning events, but such forecasts are less common. Forecast systems that are capable of predicting events with probabilities different from the Prevalence frequency are said to have 'sharpness'−and our forecasts thus exhibit sharpness.

Having demonstrated that FlashNet is reliable and much more skillful than a simple Prevalence-based random model, in the next section we compare FlashNet's skills against those of HRES in predicting lightning flash occurrence.

### FlashNet against HRES: skills evaluation

No well-documented attempts to exploit NWP models and AI in concert for predicting lightning flashes in the forecast horizon up to several days ahead seem to be present in the current peer-reviewed

**Table 1 | The Precision and Recall scores computed from the HRES 'litota3' variable are reported for the two forecast intervals 0–24 and 24–48 h**

|  | 0–24 h | | | | | | 24–48 h | | | | | |
|---|---|---|---|---|---|---|---|---|---|---|---|---|
|  | Balanced | | | Unbalanced | | | Balanced | | | Unbalanced | | |
|  | 2019 | 2020 | 2021 | 2019 | 2020 | 2021 | 2019 | 2020 | 2021 | 2019 | 2020 | 2021 |
| Precision | 0.92 | 0.91 | 0.90 | 0.11 | 0.10 | 0.09 | 0.92 | 0.91 | 0.90 | 0.11 | 0.09 | 0.08 |
| Recall | 0.72 | 0.74 | 0.67 | 0.72 | 0.74 | 0.67 | 0.67 | 0.70 | 0.62 | 0.67 | 0.70 | 0.62 |

Results are reported for both the balanced and unbalanced datasets, and separately for the years 2019, 2020, and 2021.

literature. We came across an attempt documented in conference proceedings[37] which, however, does not allow for detailed information extraction. This limits the possibility of doing comparisons of our strategy with other AI-enhanced forecast approaches. We can however compare part of our results with the deterministic predictions from the world-leading forecast model HRES operational at ECMWF.

Precision and Recall scores can be easily computed from HRES through the 'litota3' variable[38] (see the "Methods" section). The resulting scores are summarized in Table 1 for the three folds of the 3-fold cross-validation in the 0–24 and 24–48 h forecast intervals, for both balanced and, actual, unbalanced datasets. For the sake of clarity, to calculate the Precision in the 0–24 h forecast interval, the 'true positive', TP, of all 3-h sub-interval forecast intervals are summed up to define the total number of TP in the 0–24 h forecast interval. The same holds for calculating FP and FN entering in the Precision and Recall scores (see the "Methods" section). A similar procedure holds for the 24–48 h forecast interval. While the Precision score depends on Prevalence (see again the "Methods" section), this is not for the Recall[34] score which is the same passing from the balanced to the unbalanced dataset.

Since HRES does not provide a probabilistic forecast, the P-R curve cannot be computed from HRES, making it impossible for a direct comparison against our curves. Rather than comparing those curves, we can compare the resulting FlashNet's Recall once the Precision score is fixed to the value of HRES. It is indeed worth remembering that FlashNet is a fully probabilistic network and by varying the probabilistic threshold of lightning detection we can easily set our model Precision to coincide with the one of HRES. Why we do not follow the opposite option (i.e. fixing the model Recall at the one of HRES) is related to the highly sensitive character of our prediction. Failing to predict the occurrence of lightning can indeed cause more serious consequences than erroneously predicting its occurrence. Minimizing false negative events (FN) is thus of imperative importance in the forecast problem at hand.

Figure 4a shows the resulting FlashNet's Recall score (blue curve) together with the one from HRES (orange curve). Both scores are plotted as a function of the forecast lead time, up to 48 h. The test set corresponds to the whole years 2019, 2020 and 2021 of the exploited 3-fold cross-validation, on the whole area of Fig. 1. It is a huge test set formed by a total of about 70 million prediction–observation couples. The insets highlight the added value brought out by FlashNet in terms of FlashNet's Recall skill score (see the "Methods" section) using HRES as a reference.

Several remarks from Fig. 4a are worth discussing. It is remarkable that FlashNet reaches values of the Recall score up to 95% against the 85% of HRES. The higher Recall possessed by FlashNet is however systematic, being valid for each forecast lead time and each fold of the cross-validation. Passing from the 0–24 h to the 24–48 h forecast interval, as for HRES, also for FlashNet a reduction of the skills can be observed. However, FlashNet in the 24–48 h forecast interval is not only superior to HRES in the corresponding interval, but it also overperforms HRES in the 0–24 h interval. Interestingly, the skills of HRES display a clear dependence on the diurnal cycle, with poorer skills

during nighttime hours compared to daytime hours. This phenomenon is well-documented in the literature[39] and is attributed to insufficient nighttime convection, a known shortcoming in IFS cycle 47r2 forecasts of convective activity.

Worth remarking is also the behavior similarity between FlashNet's and HRES' Recall curves. This is however not surprising: FlashNet indeed uses the features from HRES (except for the variable 'litota3') and thus its skills are intrinsically linked to those of HRES (see again Fig. 2b) with its peaks qualitatively reflecting those of HRES in Fig. 4a. If lightning proxies were poorly predicted by HRES, there would be no way for FlashNet to learn the mapping between HRES features and lightning occurrences. From this perspective, having accurate predictions of the lightning proxies forecasted by HRES is a necessary (but not sufficient) condition for the good predictive performance of FlashNet. The skills of our network are thus an indirect way to assess the skills of HRES in relation to meteorological variables triggering the lightning occurrence.

Our network is however appreciably more skillful than the HRES algorithm in mapping the skillful meteorological features from HRES into a dichotomous index for detecting lightning occurrence. Finally, from the insets of Fig. 4a, we only note a slight reduction of the resulting Recall skill scores passing from the 0–24 h to the 24–48 h forecast interval. This tells us that learning the mapping between meteorological features and lightning occurrence can be a successful task even for longer temporal horizons. Although the absolute skill in predicting lightning occurrence decreases with the forecast lead time, the added value brought by FlashNet persists. Table 2 summarizes the resulting Recall scores from FlashNet for the three folds of the 3-fold cross-validation in the 0–24 and 24–48 h forecast intervals, for both balanced and, actual, unbalanced datasets.

We now focus on evaluating the performance of FlashNet and HRES under different prediction conditions over the sea and land. The convection mechanisms in these regions are indeed different, and it is of interest to examine whether the model's skills vary. It should be noted that the training of FlashNet was conducted without distinguishing between sea and land; the differentiation is made only during the testing phase. The predictive capacity of FlashNet can only further improve by conducting separate training for sea and land. In Fig. 4b, we present the Recall (with FlashNet's Precision fixed to that of HRES) for FlashNet and HRES over the sea (dashed lines) and over land (solid lines) in two forecast intervals: 0–24 and 24–48 h (upper panels). The lower panels depict similar curves but for Precision, with FlashNet's Precision determined by fixing the Recall to the value of HRES. In all cases, we fused the three folds and analyzed the indices month by month. Several comments are worth discussing. The most evident is that both over sea and land, FlashNet outperforms HRES, both in terms of Precision and Recall. The reason behind the lower skill scores observed for the Precision index can once again be explained through Eq. (8) and, ultimately, by the low prevalence of the dataset. Another interesting aspect is the more pronounced improvement brought by FlashNet on land compared to over the sea. This aspect does not seem surprising: predicting convection triggered by orographic effects is clearly challenging for a model like HRES, which has a resolution of

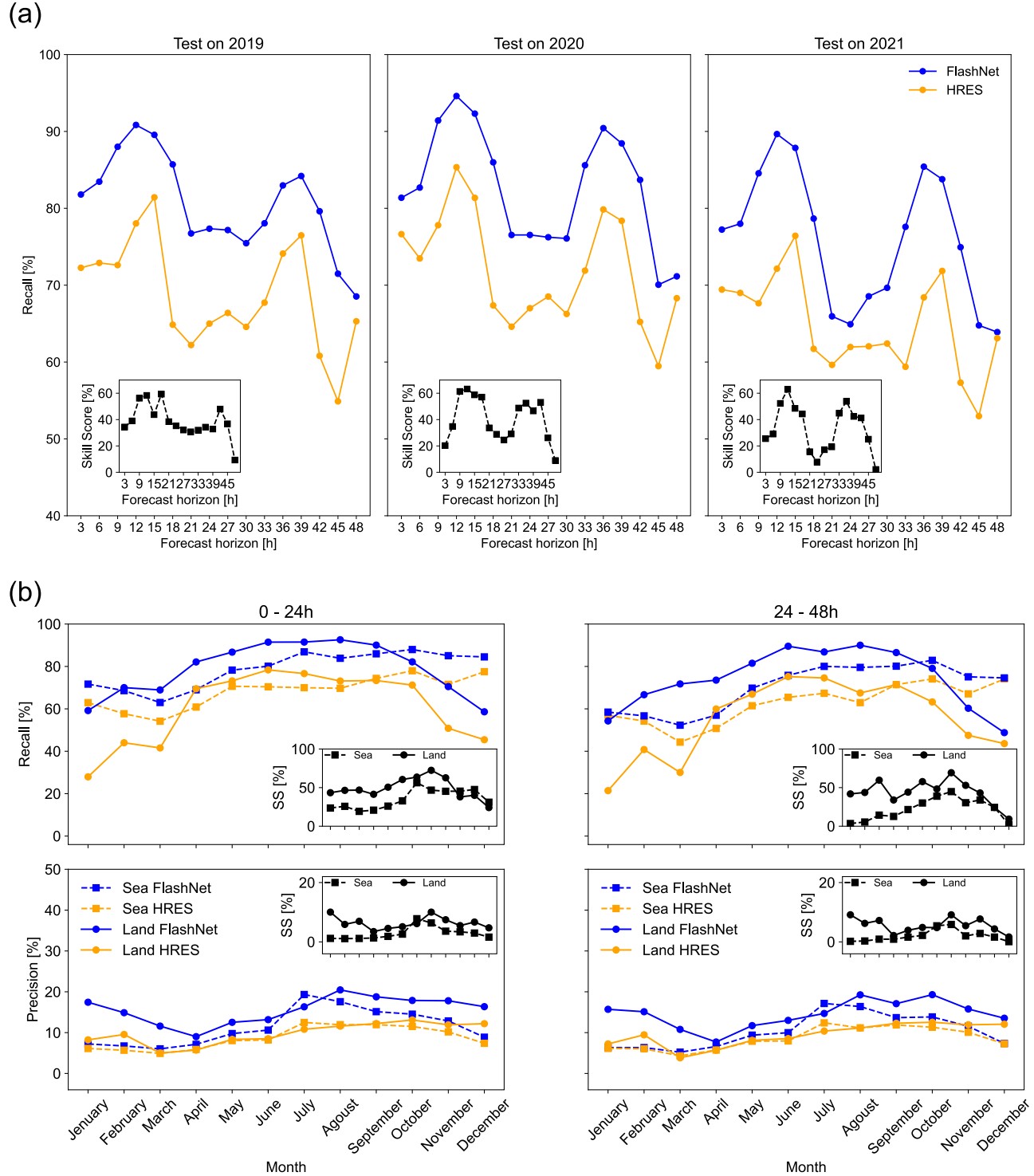

**Fig. 4 | Assessing FlashNet's predictive skills against the ECMWF-HRES model.**
**a** The Recall score vs. the forecast lead time from FlashNet (blue curve) and from HRES (orange curve). From left to right, the 3 test sets. They correspond to the 3 (naturally unbalanced) years 2019, 2020, and 2021 of the 3-fold cross-validation. Insets: FlashNet's Recall skill scores using HRES' Recall as reference. **b** The Recall, with FlashNet's Precision fixed to that of HRES, for FlashNet and HRES over the sea (dashed lines) and over land (solid lines) in two forecast intervals, 0–24 and 24–48 h (upper panels), are shown. The lower panels depict similar curves but for Precision, with FlashNet's Precision determined by fixing the Recall to the value of HRES. In all cases, we fused the three folds and analyzed the indices month by month. Source data are provided as a Source Data file.

approximately 10 km in the area of interest. In contrast, FlashNet learns these effects from the observed data during the training period.

Regarding the trends of these indices in different months of the year, there are no particular trends observed, except for a slight tendency (highlighted by the skill scores reported in the insets) towards increased predictive skills of FlashNet compared to HRES in the summer/autumn months. This seems to hold true both over the sea and over land.

**Table 2 | For each fold of the 3-fold cross-validation, shown are the Recall scores in the two forecast intervals 0–24 and 24–48 h, from FlashNet and HRES**

| | 0–24 h | | | | | | 24–48 h | | | | | |
|---|---|---|---|---|---|---|---|---|---|---|---|---|
| | Balanced | | | Unbalanced | | | Balanced | | | Unbalanced | | |
| | 2019 | 2020 | 2021 | 2019 | 2020 | 2021 | 2019 | 2020 | 2021 | 2019 | 2020 | 2021 |
| FlashNet | 0.86 | 0.88 | 0.81 | 0.86 | 0.88 | 0.81 | 0.78 | 0.82 | 0.76 | 0.79 | 0.82 | 0.76 |
| HRES | 0.72 | 0.74 | 0.67 | 0.72 | 0.74 | 0.67 | 0.67 | 0.70 | 0.62 | 0.67 | 0.70 | 0.62 |

Results for both the balanced and the unbalanced test sets are displayed. The probability threshold is fixed in the FlashNet network in a way that the resulting Precision equals the one from HRES reported in Table 1.

**Table 3 | The mean difference between observed and predicted lightning occurrence (0 and 1 representing no lightning occurrence in 1 h while 1 denotes the occurrence of at least one lightning flash in 1 h) by HRES and FlashNet (BIAS in short) is reported**

| | 0–24 h | | | | | | 24–48 h | | | | | |
|---|---|---|---|---|---|---|---|---|---|---|---|---|
| | 2019 | | 2020 | | 2021 | | 2019 | | 2020 | | 2021 | |
| | F1 | Bias | F1 | Bias | F1 | Bias | F1 | Bias | F1 | Bias | F1 | Bias |
| FlashNet | 0.37 | −0.005 | 0.38 | −0.013 | 0.36 | −0.005 | 0.37 | −0.008 | 0.38 | −0.017 | 0.36 | −0.008 |
| HRES | 0.19 | −0.066 | 0.17 | −0.056 | 0.15 | −0.059 | 0.18 | −0.062 | 0.17 | −0.054 | 0.15 | −0.055 |

Also reported is the F1 score from HRES and FlashNet.

In Table 3, we have shown the skill with which the HRES model and our FlashNet network replicate the Prevalence of the observed dataset. To achieve this, we calculated, for the forecast intervals of 0–24 and 24–48 h, the mean difference between observed and predicted lightning occurrence (i.e., sequence of 0s and 1s) by HRES and FlashNet. This observable measures the disparity between the Prevalence of the observed dataset and the one reconstructed from the forecasting models (HRES or FlashNet). In Table 3, we will refer to this difference as BIAS for brevity. The table also shows the F1 score (see the "Methods" section) for both HRES and FlashNet. For the latter model, the F1 score has been minimized in the validation set by selecting the optimal probability decision threshold. The resulting sequences of 0s and 1s are determined from FlashNet based on this threshold, with 1 being selected when the predicted probability crosses the threshold, and 0 being assigned when the predicted probability is below the same threshold. FlashNet exhibits a clear superiority over HRES in terms of both BIAS and F1 score across all years and forecast intervals.

Up to this point, we have assessed the performance of both HRES and FlashNet using highly quantitative error metrics. We conclude this section with a qualitative comparison that provides an overall view of FlashNet's superiority over HRES. To achieve this, we fused the data from the three test folds and accumulated all the resulting 0s (no lightning) and 1s (occurrence of at least one lightning event) for each trimester of the resulting 'typical' year. A given integer value in the maps thus tells us how many events containing at least one lightning flash in 1 hour occur. For FlashNet, the sequences of 0s and 1s have been determined by selecting the probability decision threshold which minimizes the F1 score in the validation set. In Fig. 5 we present the accumulated 0/1 values across the entire analyzed domain. This aggregation step is essential due to the rarity of lightning occurrences, which represent <1% of the total events. For each trimester, we display three maps (one for HRES, one for FlashNet, and one for observations) side by side, facilitating visual comparison. This comprehensive view seems to qualitatively affirm what all the quantitative statistical metrics we employed had previously established: FlashNet's superiority over HRES in predicting lightning occurrences.

## Discussion

A hybrid AI-enhanced methodology, which we have dubbed FlashNet, has been presented and tested to predict the occurrence of lightning flashes in the 0–48 h forecast horizon. Our method is not purely data-driven; instead, features from the ECMWF world-leading NWP model HRES are mapped onto the probability of lightning flash occurrence via a training set where the truth corresponds to the lightning occurrence from the LAMPINET detection system covering the whole Italian territory. From our results, the superiority of FlashNet predictions over HRES' predictions based on the fully deterministic paradigm clearly emerges. FlashNet systematically outperforms HRES in predicting lightning occurrence, the conclusion being valid for each forecast lead time and each fold of the cross-validation. Rather impressive is the peak achieved for the Recall, of about 95% against the 85% of HRES, occurring around noon in the 0–24 h forecast interval.

Also remarkable is the fact that FlashNet in the 24–48 h forecast interval not only overcomes HRES in the corresponding interval, being also superior when the comparison is done with respect to the HRES' Recall in the 0–24 h interval. Our results suggest that learning the mapping between meteorological features and lightning occurrence might be a successful task even for forecast horizons longer than the 0–48 h considered in the present study. Although the absolute skill in predicting lightning occurrence is expected to decrease with the forecast horizon, the added value brought by FlashNet would persist as observed in the present study passing from the 0–24 h to the 24–48 h forecast interval. This aspect opens up promising avenues for future research, particularly in extending the forecast horizon beyond 48 h.

But why is FlashNet successful? A significant portion of the credit goes to HRES: as the best existing deterministic model, it provides excellent large-scale proxies for lightning. We refer, e.g., to thermodynamic and variables, cloud microphysics representation, and convection proxies. Furthermore, despite its simplicity, the FlashNet architecture has proven to be robust against overfitting, demonstrating excellent generalization of its performance from one fold to another. Conversely, HRES employs a deterministic and relatively simple lightning initiation model that is, as a result, unable to fully capture the complexity of the initiation phenomenon.

Our strategy also significantly outperforms the prediction of lightning occurrence made in terms of a simple Prevalence-based random model. Finally, FlashNet is also calibrated, and thus reliable: the resulting reliability diagram tells us that our network is "very useful for decision making" according to the criteria for classifying forecast skill into five categories based on the slope of the reliability line in the

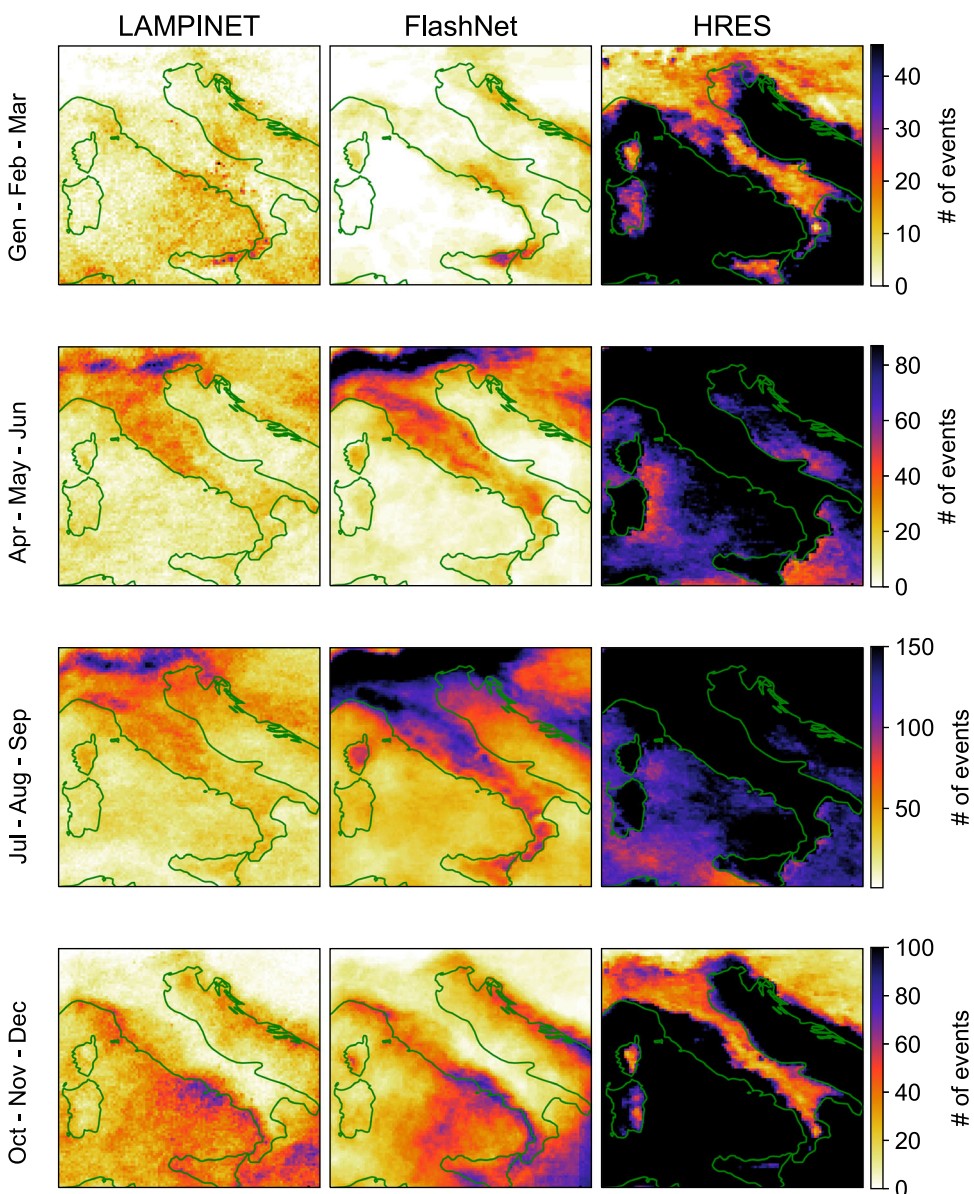

**Fig. 5 | Spatial distribution of lightning flash occurrence.** The data from the three test folds have been fused and the resulting 0s (no lightning) and 1s (occurrence of at least one lightning event) accumulated point-by--point across the entire analyzed domain, for each trimester of the three considered years. For each trimester, we display the four maps for LAMPINET network (the ground truth), FlashNet, and HRES. Different colors represent different numbers of events according to the color bar reported for each panel row. The reported domains extend from 36.0°N to 47.0°N in latitude and from 7.5°E to 19.0°E in longitude.

reliability diagram[35]. The strategy we have proposed and validated can be easily generalized to other regions in the world and also using other lightning flash detection systems for training and testing. In its present form, our AI-enhanced strategy could already be used as a post-processing of the outputs operatively produced by HRES in the 0–48 h forecast horizon. Moreover, because it emerges from our study that a necessary condition for accurate predictions by FlashNet is to have accurate forecasts of proxies relevant to lightning initiation, in this perspective, having an even more accurate model than HRES could lead to overall more accurate results. Considering the success of Pangu-Weather[16] discussed in the introduction, it emerges as an ideal candidate to take this further step.

A further line of generalization for our work is to incorporate observed features alongside those extracted from model forecasts to further enhance the quality of lightning predictions, specifically for short-term temporal horizons. Among the features of interest, observations from the LAMPINET network are certainly an option to be

considered. Additionally, aerosol-related features also merit attention in order to more effectively elucidate lightning mechanisms, as demonstrated in ref. 40. In the present study, a short-term forecast refers to the initial few forecast hours (e.g., 1 or 2 hours after the analysis at 00 UTC) of the night following the 00 UTC analysis. This suggests that incorporating observations might provide some benefit for the first (or at least a portion of it) forecasted night. Unfortunately, we firmly exclude the same potential benefit for the night in the 24–48 h forecast horizon (and, more generally, after the first few hours from the analysis time), as such a lead time is too distant from the observation time to maintain a relevant correlation with it, as explicitly shown by Casciaro et al.[41] in relation to wind forecasts.

Because of the generality of our approach, the present work is expected to open new avenues of research and activities in all environmental fields dealing with rare events, including floods and extreme temperatures/winds, issues becoming more and more important in a climate change scenario. Furthermore, having

demonstrated FlashNet's capability in predicting lightning flash occurrence, the challenge of predicting the actual density of lightning can be now addressed by transitioning from a classification problem to a regression-oriented approach.

## Methods

### The LAMPINET detection system

The Italian lightning network[31,32], reported in Fig. 1, is based on Vaisala technology, with 15 IMPACT ESP sensors uniformly distributed over the national territory, and operative since 2004. It is based both on magnetic direction finding (MDF) and Time Of Arrival (TOA) techniques. LAMPINET network can reach a detection efficiency of 90% for intensity ($I$) > 50 kA, and location accuracy of 500 m over the whole Italian territory[31], thus much higher than the HRES' spatial resolution. Since 2021, LAMPINET sensors have become part of the European EUCLID network[42].

### The global ECMWF HRES model

The HRES model is currently run by ECMWF[12] with outputs organized on a constant 0.1° × 0.1° lat/lon grid, corresponding to about 10 km resolution, horizontally, in the considered area of Fig. 1. For the present study, forecast data relative to the entire Italian territory and for the same time span of the observed data from LAMPINET are collected every 3 h. The ECMWF conducts four daily runs of the HRES model. These runs start from the so-called 'analysis' (i.e. the initial conditions) at the synoptic hours 00:00 UTC, 06:00 UTC, 12:00 UTC, and 18:00 UTC. For our experiments, we have chosen the 00:00 UTC model run, since it plays a pivotal role in shaping early morning and daytime weather predictions, making it of paramount importance in the operational weather model's daily cycle. HRES provides a considerable number of meteorological variables, both two-dimensional (either surface fields or fields integrated along the column) and fully three-dimensional fields varying on multiple pressure levels, constituting the predicted state of the atmosphere up to 10 days ahead. The features we have collected to train FlashNet are listed in Table 4. For each model grid point, we extracted 9 two-dimensional fields and 5 vertical profiles defined on 7 pressure levels. Moreover, from the extracted variables we derived other variables (see again Table 4).

For the training phase of the neural network, the selected HRES gridded data sampled every 3 h and belonging to a domain extended from 36.0°N to 47.0°N in latitude and from 7.5°E to 19.0°E in longitude, for two different forecast intervals, 0–24 and 24–48 h, have been considered. Among other meteorological variables provided by HRES, worthy of attention is a family of variables encoded with the root 'litota'[38]. These variables, available since mid-2018, account for the averaged total (cloud-to-cloud and cloud-to-ground) lightning flash density in different time intervals. In the HRES model, the total lightning flash density is calculated using an empirical formula involving convective cloud and precipitation information, convective available potential energy (CAPE), and convective cloud base height diagnosed by the convection scheme[43]. The HRES lightning flash forecast provided by 'litota3', i.e. the averaged total lightning flash density in 3 h, is used to obtain a benchmark of FlashNet's predictions (see next section). It is worth noting that 'litota3' (or other variables with the root 'litota') is not used as a feature to train our neural network.

### Lightning dataset creation

In the present study, the complete LAMPINET archive of the lightning flash detection recorded from 1 January 2019 to 31 December 2021 has been used. Raw LAMPINET data contains a continuous-in-time mapping (in terms of lat-lon coordinates) of lightning activity across the geographic area of Fig. 1, including information on the type of lighting

**Table 4 | Features used to train FlashNet**

| |
|---|
| **List of gridded features** |
| CAPE index (convective available potential energy) |
| cp—convective precipitation |
| tcw—total cloud water |
| tp—total precipitation |
| mslp—mean sea level pressure |
| hcc—high cloud cover |
| sp—surface pressure |
| $w_{925}$—vertical velocity at 925 hPa |
| $w_{850}$—vertical velocity at 850 hPa |
| **List of derived gridded features** |
| capetp—product between CAPE and tp |
| $\Delta t1$—temperature difference between 925 and 850 hPa |
| $\Delta t2$—temperature difference between 850 and 700 hPa |
| $\Delta t3$—temperature difference between 700 and 600 hPa |
| $\Delta t4$—temperature difference between 600 and 500 hPa |
| $\Delta t5$—temperature difference between 500 and 400 hPa |
| $\Delta t6$—temperature difference between 400 and 300 hPa |
| $\Delta q1$—specific humidity difference between 925 and 850 hPa |
| $\Delta q2$—specific humidity difference between 850 and 700 hPa |
| $\Delta q3$—specific humidity difference between 700 and 600 hPa |
| $\Delta q4$—specific humidity difference between 600 and 500 hPa |
| $\Delta q5$—specific humidity difference between 500 and 400 hPa |
| $\Delta q6$—specific humidity difference between 400 and 300 hPa |
| $\Delta\theta1$—potential temperature difference between 925 and 850 hPa |
| $\Delta\theta2$—potential temperature difference between 850 and 700 hPa |
| $\Delta\theta3$—potential temperature difference between 700 and 600 hPa |
| $\Delta\theta4$—potential temperature difference between 600 and 500 hPa |
| $\Delta\theta5$—potential temperature difference between 500 and 400 hPa |
| $\Delta\theta6$—potential temperature difference between 400 and 300 hPa |
| **List of gridded features corresponding to vertical profiles** |
| q—specific humidity |
| gh—geopotential height |
| t—temperature |
| u—wind speed eastward component |
| v—wind speed northward component |
| **List of derived features from vertical profiles** |
| uq—eastward component of the q-vector |
| vq—northward component of the q-vector |
| $\theta$—potential temperature |
| grad$\theta$—vertical gradient of potential temperature |
| pv—potential vorticity |
| **Other features** |
| $\cos(2\pi H/24)$ —H being the hour of the day |
| $\sin(2\pi H/24)$ —H being the hour of the day |
| $\cos(2\pi m/12)$ —m being the month of the year |
| $\sin(2\pi m/12)$ —m being the month of the year |
| lat—latitude |
| lon—longitude |
| z—surface geopotential |

For all gridded features, we also calculated the mean, maximum, minimum, and standard deviation over a region of 11×11 grid points around the reference HRES' grid point. The features extracted in the form of vertical profiles refer to the pressure levels 925, 850, 700, 600, 500, 400, and 300 hPa.

(cloud-to-cloud or cloud-to-ground) and their amperage. These information have been aggregated in time, on time intervals of three hours, and in space, by arranging them on the same grid on which the HRES' output is available. In plain words, for each HRES' grid point, we counted the number of lightning flashes that occurred in a region of an area of 10 km × 10 km around the grid point. Each map obtained in this way has then been converted into a map of '0' and '1', the former being associated with the absence of lightning in 3 h in the 10 km × 10 km area around a given grid point, the latter telling us that at least one lightning event occurred in three hours in the same area around the grid point. A similar dataset of '0' and '1' has been created starting from the 'litota3' HRES's variable[38].

The resulting dataset is strongly unbalanced with only about 1% of events. Using a similar dataset for the learning stage of our AI-based strategy is challenging. The AI network indeed sees too few 1-type events, compared to the 0-type events, to learn the mapping of HRES' features onto the probability of lighting flash occurrence without biases. To overcome this problem, different techniques have been exploited here, such as "oversampling"[44], "undersampling"[45], "synthetic minority oversampling technique" (SMOTE)[46]. We provide details solely on the undersampling strategy which has been the one giving the best results. Accordingly, our resulting balanced dataset contains all lightning events while picking at random in the whole original dataset a number of 0-type events in a way that the final number of these latter equals the number of lightning occurrences. The resulting LAMPINET balanced dataset contains for the years 2019, 2020, and 2021 a total number of about 2 million samples, all years contributing almost equally to this number.

## k-fold cross-validation

Having created a balanced dataset of lightning occurrence, we have performed k-fold cross-validation (with k = 3) to train/test the network. Accordingly, each of the 3 years (2019, 2020, and 2021) is sequentially taken as the test set with the remaining two serving for training. Variability and robustness of FlashNet's skills can be thus assessed in this way. While the training set in the k-fold cross-validation is always balanced, the test has been carried out both in its balanced and in its actual unbalanced form (i.e. testing the entire year with its natural degree of unbalance).

## The FlashNet deep learning framework

Figure 6 shows a schematic representation of the ensemble deep learning framework we propose here to predict lightning occurrence in the 0–48 h forecast horizons, starting from meteorological features from the ECMWF-HRES weather model. Figure 6a reports the network architecture constituting the single ensemble member. To properly deal with both local and nonlocal features (i.e. accounted by vertical profiles other than by point-wise meteorological information), a multi-head structure is exploited where one head is based on a neural network composed of four fully connected hidden layers. The input of the layer dealing with point-wise data is made by different gridded data provided by HRES at a given forecast lead time. Moreover, for each feature, we calculate the mean, standard deviation, maximum, and minimum value over an area of 11 × 11 grid cells around the selected grid point, which allows us to take into account large-scale spatial effects. The head dealing with non-local features is based on one-dimensional convolutions which take as input vertical profiles built from seven different HRES' pressure levels at a given point and forecast lead time. The selected variables are reported in Table 4. The final layers of the two heads are concatenated, followed by two fully connected layers leading to the final node output. Since we face a binary classification, the output is provided by a single neuron that exploits the sigmoid activation function in order to get a value between 0 and 1 which represents the probability associated with the lighting flash occurrence.

To train FlashNet we adopt an ensemble strategy. The idea is to combine multiple neural networks, where each network is trained with bootstrap replicas with a replacement of the original dataset. In addition, to maximize the independence of each model of the ensemble, each ensemble member has its own standardization for the features in input to the network. Then, the final probability output is obtained by averaging the outputs of the network as sketched in Fig. 6b. The number of hidden layers and neurons for each hidden layer, the number of filters for each convolutional layer, activation functions, level of dropout, and batch size have been chosen in order to prevent the overfitting of the single member. Each member of the ensemble network has been trained with early stopping and learning rate reduction by monitoring the out-of-bag loss. FlashNet's architecture is available in the on-line repository[47]. We trained 20 members on the training set of the 0–24 h forecast horizon, and the same ensemble networks have also been used to make predictions in the 24–48 h forecast horizon. Since we adopt an undersampling strategy to train FlashNet, there is a modification of the distribution of classes in the training set, which consequently biases the posterior probabilities of a classifier[48], although the bias due to undersampling does not affect the ranking order returned by the posterior probability. For this reason, at the FlashNet's output (see Fig. 6b), we use the Bayes minimum risk method (BMR) described in[49] in order to get reliable output probabilities.

**Neural network details.** Each member of the ensemble (see Fig. 6b) shares the same architecture, as depicted in Fig. 6a. The architecture comprises two heads. The head that takes point-wise data as input is followed by dropout, with a rate of 0.1, which acts on the 472 input features. This is followed by four fully connected layers interspersed with dropouts. We refer to the fully connected layers as $Dense_i$, with $i$ denoting the ith dense layer, as illustrated in Fig. 6a, progressing from left to right. Specifically, $Dense_1$ consists of 1416 nodes, utilizes the elu activation function, and has a dropout rate of 0.5. $Dense_2$ has 2832 nodes, uses the relu activation function, and has a dropout rate of 0.6. $Dense_3$ also has 2832 nodes, utilizes the elu activation function, and has a dropout rate of 0.6. Finally, $Dense_4$ consists of 500 nodes and uses the swish activation function.

The head that takes vertical profiles (10 profiles on 7 pressure levels) as input is based on one-dimensional convolutions. In this case, as well, the input is followed by dropout with a rate of 0.1. This is followed by $Conv1D_1$ consisting of two one-dimensional convolutions using 40 filters with a kernel size of 3 and padding set to the same. The elu activation function is applied, followed by MaxPooling1D with a pool size of 2 and dropout with a rate of 0.5. This is followed by $Conv1D_2$, which consists of two one-dimensional convolutions using 80 filters with a kernel size of 3, padding set to same, elu activation function, MaxPooling1D with a pool size of 2, and dropout with a rate of 0.55. Finally, a flattening operation is applied to $Conv1D_2$, followed by a fully connected layer consisting of 500 nodes with the swish activation function ($Dense_5$).

The two heads are linked by concatenating the layers $Dense_4$ and $Dense_5$. After concatenation, a dropout with a rate of 0.5 is applied. This is followed by a fully connected layer ($Dense_6$) with 500 nodes, the swish activation function, and a dropout rate of 0.2. Finally, the last layer, $Dense_7$, consists of 250 nodes using the relu activation function. The output node applies the sigmoid activation function and L2 regularization with default values provided by TensorFlow. Each network in the ensemble is trained using the Adam optimizer with a learning rate of 0.0001. The loss function employed is the Brier score:

$$BS = \frac{1}{N} \sum_{i=1}^{N} (p_i - o_i)^2 \qquad (1)$$

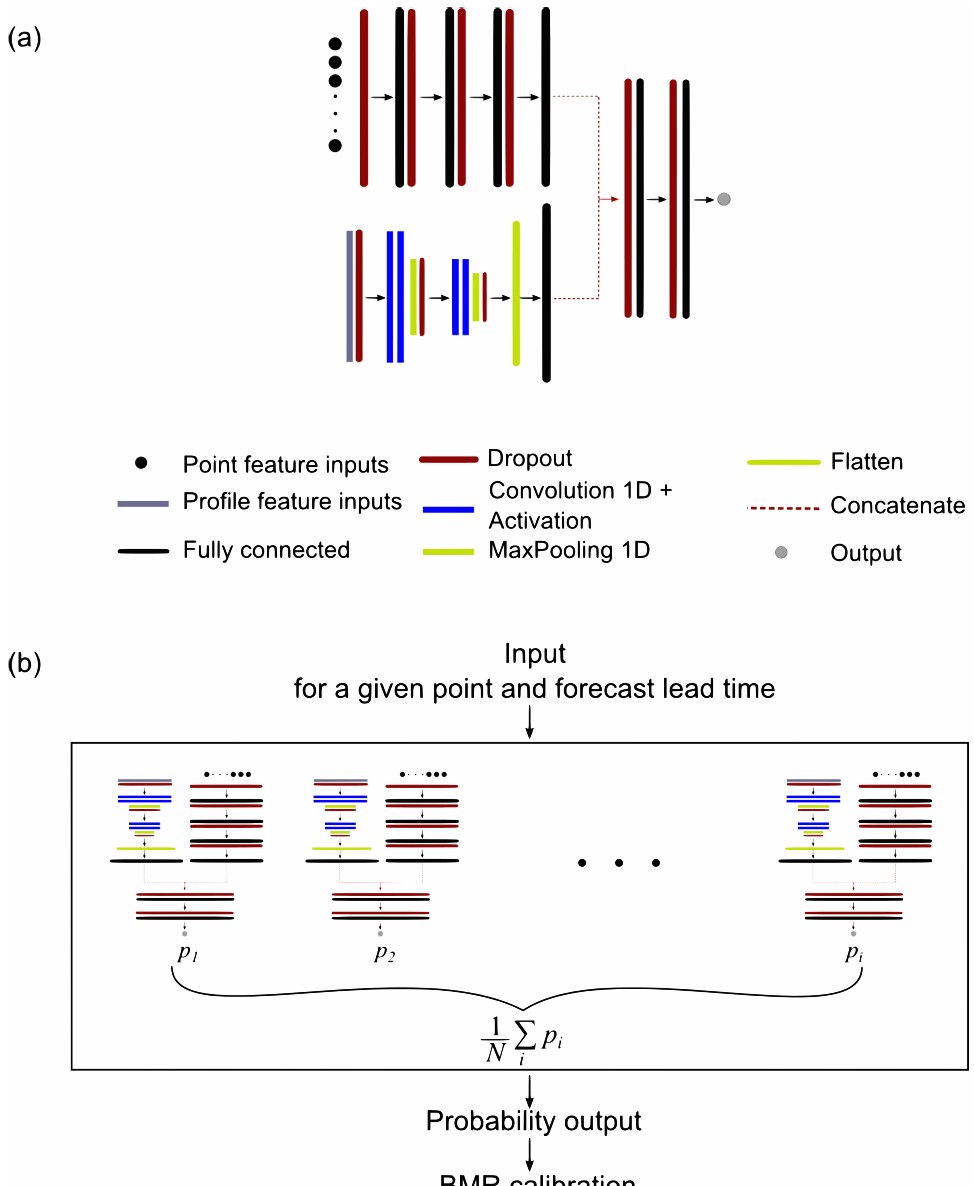

**Fig. 6 | A schematic representation of the FlashNet ensemble network.** Panel **a** refers to a single multi-head member of the ensemble. Panel **b** illustrates how each member contributes to the final outcome corresponding to the prediction of the lightning flash occurrence probability.

Here, $p$ represents the predicted probability, and $o$ corresponds to the observed 0/1 label. For each member, training is performed on boot-strap replicas with replacement of the original dataset, and each member has its own standardization for the input features. Additionally, we trained the networks using early stopping with a patience of 15 epochs and learning rate reduction with a patience of 10 epochs, employing a reducing factor of 0.1. The out-of-bag loss threshold for learning rate reduction is set at 0.0001. The hyperparameters of the network have not been individually fine-tuned, as such an approach would have been prohibitively resource-intensive. Instead, these hyperparameters, as well as architecture settings, have been manually selected, aiming to discover configurations that would reliably yield a high degree of generalization across training and validation datasets.

**Statistical indices to assess forecast skills**
**Precision and Recall scores.** As customary when dealing with unba-lanced datasets, the model skill evaluation is built in terms of the scores 'Precision', [Eq. (2)], related to the false alarm ratio (FAR) by the relation Precision = 1−FAR, and 'Recall', also known as sensitivity or probability of detection, POD, [Eq. (3)][50,51], here denoted by $P$ and $R$, respectively:

$$P = \frac{\text{TP}}{\text{TP} + \text{FP}} \tag{2}$$

$$R = \frac{TP}{TP + FN} \tag{3}$$

In Eqs. (2) and (3), TP refers to the true positives (i.e. an outcome where the model correctly predicts the event occurrence), FP refers to the false positives (i.e. an outcome where the model incorrectly predicts the event occurrence), and FN refers to the false negatives (i.e. an outcome where the model incorrectly predicts the negative class corresponding to the absence of the event). Of course, an optimal prediction would have $P = R = 1$. To identify TP, FP, and FN a threshold from 0 to 1 must be set in order to define event '1' (occurrence of lightning) and event '0' (no lightning occurrence), starting from the event probability outputted from FlashNet (see again Fig. 6). One

additional score have been employed to assess the model's classification performance: the F1 score, which is the harmonic mean of Precision and Recall and is defined as $F1 = 2PR/(P+R)$.

**The area under the precision–recall curve (AUC).** Once the threshold in the probability outputted from FlashNet is varied from 0 to 1 to identify a set of precision–recall couples, these latter can be arranged in the so-called Precision (along the ordinate)–recall (along the abscissa) plot, whose integral is the AUC score. AUC is a metric used to evaluate the performance of binary classification models, particularly when dealing with imbalanced datasets where one class is much more prevalent than the other[52]. Unlike the area under the ROC curve (AUC-ROC), which uses the receiver operating characteristic curve[53], the precision–recall AUC uses the precision–recall curve to assess the model's ability to correctly classify positive instances while considering precision and recall trade-offs.

A perfect classifier with maximum precision and recall would have an AUC of 1. A random classifier that makes predictions without considering the actual class distribution would have an AUC close to the proportion of positive instances in the dataset (i.e. the Prevalence, see next subsection). An AUC value between 0 and 1 indicates the model's ability to achieve a balance between Precision and Recall.

The precision–recall AUC score is calculated using the built-in function in scikit-learn python-based library[54], taking the precision and recall values as arguments.

**Prevalence-based random model.** *Prevalence*, denoted here by $\wp$, is defined as the ratio of the number of positive observed events, $N_1$, to the total number of events, $N_1 + N_0$. Namely, $\wp = N_1/(N_1 + N_0)$. Given the Prevalence obtained from the observations and given the threshold $P_{th} \in (0, 1)$ to identify the event '1' via a random generator uniform in the interval $(0, 1)$, it is easy to realize that the resulting probability of lightning occurrence would get $P = N_1/(N_1 + N_0) = \wp$ and $R = P_{th}$. Also, the resulting AUC from the random model is simply AUC $= \wp$. These simple predictions are useful benchmarks to assess the added value brought by more complex prediction strategies as the one presented here.

**Explicit formula for the dependence of precision score on Prevalence.** The formula we are going to derive allows a quantitative comparison between precision scores which refer to datasets of observations having different Prevalence. The problem of cross-dataset comparisons of precision being of general interest, especially to evaluate clinical tests[55], its derivation deserves a dedicated section.

Let us imagine to have a dataset with $N_1$ '1' events (lightning occurrence here) and $N_0$ '0' events (no lightning occurrence), corresponding to a Prevalence $\wp = N_1/(N_1 + N_0)$ (e.g., $\wp \ll 1$ corresponding to an unbalanced dataset). Let us now imagine to construct another dataset containing the same $N_1$ '1' events while picking at random '0' events from the previous dataset until a certain $\widetilde{N}_0$ is obtained. The new dataset has a Prevalence $\widetilde{\wp} = N_1/(N_1 + \widetilde{N}_0)$ (e.g., $\widetilde{\wp} = 1/2$, corresponding to a balanced dataset).

Our aim is to determine how the Precision score, $P$, changes by passing from $\wp$ to $\widetilde{\wp}$ (i.e. by reducing the number of '0' events from $N_0$ to $\widetilde{N}_0 = \alpha N_0, \alpha < 1$). In doing that, it is easy to realize that TP cannot vary from one dataset to the other ($N_1$ indeed remains unchanged) while $\widetilde{FP}$, the false positive in the new dataset, must proportionally reduce (there are indeed less '0' and thus it is more difficult to fail to predict type-1 events). Namely,

$$\widetilde{FP} = \frac{FP}{N_0}\widetilde{N}_0 = \alpha FP \qquad (4)$$

The resulting Precision, $\widetilde{P}$, thus reads: $\widetilde{P} = TP/(TP + \alpha FP)$ from which, after simple manipulation, one gets:

$$\widetilde{P} = \frac{P}{\alpha + (1 - \alpha)P} \qquad (5)$$

We now need to express $\alpha$ in terms of $\wp = N_1/(N_1 + N_0)$ and $\widetilde{\wp} = N_1/(N_1 + \widetilde{N}_0)$. Simple algebra leads to the following relationship:

$$\widetilde{\wp} = \frac{1}{\frac{1}{\wp} + \left(\frac{1}{\wp} - 1\right)(\alpha - 1)} \qquad (6)$$

from which we obtain:

$$\alpha - 1 = \frac{\wp - \widetilde{\wp}}{\widetilde{\wp}(1 - \wp)} \qquad (7)$$

Plugging $\alpha$ from Eq. (7) into Eq. (5) we get the final result:

$$\widetilde{P} = \frac{P}{1 + \frac{\wp - \widetilde{\wp}}{\widetilde{\wp}(1-\wp)}(1 - P)} \qquad (8)$$

It is easy to verify that this expression holds in general, irrespective of the way one follows to change the dataset unbalance.

**Assessing reliability.** Let us now pass to discuss how we assessed the reliability of FlashNet's forecasts. The concept of reliability is related to the one of calibration, meaning that a reliable forecast denotes the goodness of calibration[56]. Calibration pertains to the agreement between a forecaster's predictions and the actual observed relative frequency of a given phenomenon[57]. Focusing on the problem at hand, a forecast is perfectly calibrated if it predicts a set of cases with, say, $x\%$ probability of being a lightning event, and the frequency of lightning events contained in that set is equal to $x\%$.

To assess calibration, for each test set of the 3-fold cross-validation we have computed the probabilities of having a lightning event and we have successively partitioned the probabilities of having a lightning event into a number of 9 subsets in which each subset represents a disjoint interval of probabilities between 0 and 1. For each subset, we computed the relative frequency of examples corresponding to lightning occurrence (fraction of positive events in short) and plotted it vs. the computed relative frequency of lightning events (mean predicted probability in short). These are the instructions we followed to construct the reliability diagram.

**The skill score index.** To highlight the predictive skills of a forecast model against a reference, we resort to the well-known skill-score index[58,59]. The latter index is defined as

$$SS = \frac{a - a_{ref}}{a_{opt} - a_{ref}} \qquad (9)$$

where $a$ is an error index to assess the forecast quality, $a_{ref}$ is the error-index associated with a reference forecast, and $a_{opt}$ refers to the index value corresponding to optimality. In the present study, $a$ is the Recall score from FlashNet, $a_{ref}$ is the Recall score from HRES, and $a_{opt} = 1$, corresponds to the optimal Recall.

**Reporting summary**
Further information on research design is available in the Nature Portfolio Reporting Summary linked to this article.

## Data availability
For training and testing FlashNet, we downloaded a subset of the HRES dataset, totaling ~176 GB, from the ECMWF forecast archive catalog,

which is accessible at https://apps.ecmwf.int/archive-catalogue/?class=od&stream=oper&expver=1. Additionally, we acquired the LAMPINET data from the Italian Air Force Meteorological Service (Servizio Meteorologico dell'Aeronautica Militare). These data are freely available for research purposes upon request to the Italian Air Force Meteorological Service via email at dati.meteo@aeronautica.difesa.it. The data volume for the area considered in our study is ~250 Mb per year. We have prepared a smaller dataset by post-processing the original LAMPINET data and combining them with HRES' output. This dataset consists of samples taken every 3 h within a domain spanning from 40.7°N to 42.0°N in latitude and from 12.1°E to 14.4°E in longitude. We have made this smaller dataset available along with the code (detailed in the next section) for the forecast horizon of 0–24 h, enabling exploratory analysis. The dataset (compressed version size of about 23 Gb) can be accessed in the on-line repository[47]. Figure Source data are provided with this paper.

## Code availability

In the on-line repository[47], FlashNet's code and the pre-trained AI-network are accessible. The code is based on TensorFlow, a Python-based library for deep learning. We also used other Python libraries, such as NumPy, SKlearn, and Matplotlib. We release both the trained ensemble and the inference code. All the details, including network architecture, modules, optimizations, and hyperparameters, are also available in the repository. The trained FlashNet is now ready for operational use, also serving as a valuable resource for researchers across diverse fields.

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

## Acknowledgements
Compagnia di San Paolo is warmly acknowledged for the financial support (project AIxtreme, called "Intelligenza Artificiale" 2022). The Italian Air Force Meteorological Service (Servizio Meteorologico dell'Aeronautica Militare) is also acknowledged for having provided us with the LAMPINET data of lightning flash detection. Useful discussions with Dr. Agostino Manzato from ARPAFVG are also acknowledged.

## Author contributions
A.M. designed research; A.M., F.C., and D.S. proposed the initial concept that M.C. refined into a working strategy; M.C. developed the final AI-enhanced framework; F.F. and M.C. analyzed data and carried out model testing and validation, A.M. and M.C. wrote the paper.

## Competing interests
The authors declare no competing interests. This work was conducted independently, and no part of the research was influenced by any funding agencies. The authors have no financial or personal relationships with other people or organizations that could inappropriately influence or bias the content of the paper. All data and materials support the published claims and comply with field standards.
