## [Peer Review File · Nature Communications]

Hybrid AI-enhanced lightning flash prediction in the medium-range forecast horizonREVIEWER COMMENTS

Reviewer #1 (Remarks to the Author):

This article provides an idea to predict the lightning occurrence integrating the AI technique and the numerical weather predictions. The developed FlashNet deep learning framework demonstrates good performance in predicting the lightning occurrence in two horizons, 0-24 h and 24-48 h, respectively. The key point of the paper comes from the utilization of the HRES as applicable and useful inputs representing the intermediate of numerical simulation. In addition, the three enhancements handling the AI challenge, namely the unbalanced dataset, meteorological feature prediction and unfamiliar conditions, are addressed as the advantage of the model. The article showed a successful solution to the objective of lightning nowcasting. However, the novelty needs to be carefully demonstrated and emphasize the difference to previous works which uses model simulation results as AI inputs and observations as outputs. Moreover, this paper has not provided sufficient scientific interpretations on the result which should be improved. The comments are below.

Major Comments:

1. This paper shows the prediction capability of the lightning occurrence based on the FlashNet, and provides evaluation on the model performance regarding to different horizons. The result presentation seems sound. However, a more interesting aspect for the researchers in the field would be, what scientific factor has assured the model performance in 24-48 h horizon in the scientific view (for example, Figure 3-5, Table 2). For example, what is happening in the predicting horizon on 9 and 33 hours, when both recall and PRC-AUC reach peak? Please try to tell the stories behind figures and share the significance of science.

2. It seems to me that the outstanding of the model performance is largely dependent on the balanced dataset (from Figure 2). However, the dependence on the balanced dataset would be highly variable in terms of the machine learning tools. For example, in Mostajabi et al., which is a tree-based machine learning tool, model shows comparable performance to the PRC curves in Figure 2 as their unbalanced dataset are only slightly pre-processed. In that case, the main technical contribution of this research would be the optimization of the applicability on the neural networks to predict lightning rather than the integration of AI and NWP. Please provide more discussions on this issue.

Reference:

[1] Mostajabi, A., Finney, D.L., Rubinstein, M. and Rachidi, F., 2019. Nowcasting lightning occurrence from commonly available meteorological parameters using machine learning techniques. *Npj Climate and Atmospheric Science*, 2(1), p.41.

3. The presentation of this paper is mostly good, while the section "more on previous works: two different competing paradigms of prediction" is confusing. Since the introduction part has already compared the previous studies, including AI, NWP and AI integrated with NWP, the organization would cause redundancy to the description. It would be better to highlight different aspects of the research and provides the insight into the novelty.

4. Considering the applicability of the model, it would be better if more evaluations under different scenarios are provided. In the temporal aspect, most other studies would focus on certain months of the year when lightning occurrence tends to be active. For example, Sarkar et al. and Song et al. focused on MAM and JJA respectively. It would be of high interest if the model developed in this paper can perform well through the year and achieve low FAR even when lightning occurrence is very rare (for example, winter, early spring). Also, how well the model can nowcast the lightning occurrence when lightning has not been observed for long consecutive hours? With respect to the prediction results that are unreliable, what are the features that causes the wrong prediction result?

Minor Comments:

1. I'm not sure if it is the formatting issue but the figure next to "this is the resulting reliability diagram" is not seen in the document.
2. Before the result section ,the advantages of the proposed model are impressive. However, this paper seems only rationalized the undersampling issue, while the prediction with extreme meteorological events and the unseen encounter have not been explained in the subsequent sections.
3. The explanation of the inset of Figure 4 seems confusing. Please rephrase the sentence.
4. The sentence " LAMPINET network can reach a detection efficiency of 90% for intensity (I) > 50 kA, and location accuracy of 500 meters over the whole Italian territory, thus much higher than the HRES' spatial resolution." also needs citation
5. The parameters (for example, the number of hidden layers) of the ensemble model, as shown in Figure 6(b), should be provided.
6. "The skills of our network are thus an indirect way to assess the skills of HRES in relation to meteorological variables triggering the lightning occurrence." The rationale is needed.

Reviewer #2 (Remarks to the Author):

"AI vs fully-deterministic algorithms: unraveling the dilemma for lightning prediction in the medium-range forecast horizon" by Cavaiola et al.

It is a very interesting, timely and important study. The results are convincing and well presented.

The methodology used is also well described.

The paper deserves to be published.

I have few minor comments

1. While authors have used 3 years of data set for the whole year and compared the forecast by HRES and Flasnet, it is worth to show the complete frequency distribution or probability distribution of lightning flashes from observation and from forecast(deterministic and AI)
2. Authors may provide the spatial distribution of lightning flashes as observed and predicted by the AI and HRES
3. It is also worth to mention whether there is any systematic bias in the observed data sets etc.
4. As the region chosen has coastal area as well as mountainous region, it will be worth mentioning whether the HRES and AI both equally perform well in these heterogeneous spatial region or there is some improvement by one over another

Reviewer #3 (Remarks to the Author):

This paper presents an hybrid (physics-based combined with AI-based) method for lightning prediction in medium-range, e.g. 2 days. The method is simple and contains two steps: (1) using the state-of-the-art physics-based model (i.e. the ECMWF-HRES model) for extracting meteorological features, (2) feeding these features into a trained neural network for final prediction. According to the experiments, the prediction accuracy is higher than only using the ECMWF-HRES model.

Overall, this paper is interesting and provides useful results to the real-world applications. I evaluate the paper in the following aspects.

1. This paper shows the insight that physics-based method and AI-based method shall be combined towards more accurate prediction. This is a promising direction provided that AI-based methods for weather forecasting have emerged in the past few years.

2. Despite the above insight, the proposed method has to be well argued, especially regarding the design choices and technical details. The reasonability of some aspects need to be explained.

(1) I am not sure if the authors tried to directly use the features extracted by the AI-based method as

the input for the neural network. In particular, an open-sourced model, Pangu-Weather (which the authors know about and cited in the manuscript), has surpassed the ECMWF-HRES in the reanalysis data and shown good results in IFS initial data. I wonder if this implies better meteorological features can be extracted and used for lightning prediction (as well as other applications). Although AI-based methods mostly provided features in a lower-resolution, they can be simply interpolated (or not, but using neural networks for processing) for the purpose in this paper. I urge that such experiments are performed and compared to the proposed method. By the way, I understand that these AI-based models do not hinder the contribution of the proposed method, but the contribution needs to be further justified by the mentioned experiments.

(2) The neural network architecture seems not well discussed. There are many design choices, such as the number of layers, the choice of convolution against transformers, the dimension (number of neurons) of hidden layers, and many more. Without sufficient discussion based on ablation and/or diagnosis, I am not sure about the superiority of the presented design.

3. The experiments also need some further justification.

(1) The data were collected in a limited region of the Earth and contains a small number of lightning events. I am not sure if the dataset is very easy for AI to fit and learn. Additionally, this is a highly imbalanced dataset, which is natural since lightning is kind of a rarely happening event. I am wondering if AI just learns to amend kind of bias from physics-based models (the ECMWF models were not tuned to fit lightning but aimed to offer global weather forecasting). If my hypothesis is true, this will not hinder the contribution of this work, but the paper needs further investigation and explanation on this topic.

(2) All experiments were performed in 0-24h and 24-48h, but both physics-based and AI-based methods have the ability of longer-range forecasting. It would be good to evaluate the method for lightning prediction in 3-5 days or even longer.

(3) I read the experiments carefully; it offered extensive results, but lacks analysis on why the proposed method can improve the results of ECMWF-HRES significantly. This may be closely related to (1) where a possible explanation lies in the amendment of bias, yet further analysis and discussions are needed.

4. The paper seems not well organized and elaborated.

(1) The introduction (including the second part, "more on previous works...") seems too long. It would be better to squeeze this part and leave details to the appendix.

(2) The main article contains little description about the proposed method, but simply lists experiments. I think basic descriptions about the proposed method shall be put into the main part.

(3) Important technical details and implementation details shall be put into the appendix. This includes but not limited to the description of network architecture, the length of training, how the loss function behaves, etc.

(4) There are issues about the submitted code. The paper offers a link with DOI, but a single ZIP file (23GB) is provided, and it is somewhat difficult for me to download it.

5. There are some typos in the paper. For example, in Page 3, "with different level*s* of maturity". Please check the manuscript carefully.

Reviewer #4 (Remarks to the Author):

Overall

- This is a very good paper exploring the use of AI for predicting lightning at longer lead times. It is suitable for publication in Nature Communications after Minor Revision.
- There are several instances (particularly in the “More on previous works”, “Results”, and “Measuring FlashNet’s” sections) where there are carriage returns, as if the end of the paragraph, but no tab to start the next paragraph. To me it seems that most or all of these are not intended to mark a new paragraph, but maybe a copy/paste or text encoding issue. Please fix.

Abstract

- “The prediction proficiency...” This is awkward in English – I would recommend replacing “proficiency” with “accuracy” or similar word.

Introduction

- “However, to set all free parameters, ... lightning flashes being rare events.” This sentence caused me consternation because, with 44 flashes occurring every second globally, it isn’t true that lightning flashes are rare events. I think your point is that lightning is rare in a machine learning sense, that is, a set of sample images over a region of interest will have pixels with lightning only rarely. So, I think what is missing from this sentence is that to set all free parameters “over a region of interest” ... This also gets at the concept that lightning relationships to meteorology may vary from region-to-region, which is further motivation for a machine learning based approach.
- “...focus is restricted to the sole nowcasting” This is awkward in English; I would recommend changing to “focus is restricted solely to nowcasting”.
- “When NWP models are used in synergy with AI methods, it turns out that the forecast horizon was always limited to a few hours ahead.” Why is this? The focus of the studies is on nowcasting, but I don’t see any reason why NWP/AI synergy should be limited to nowcasting. This is confusing, please reword.

More on previous works

- “Also, ERA5 can hardly been interpreted...” Grammar: change “been” to “be”.

Results

- You mention the Methods section – can this be a “Data and Methods” section instead? If so, please make that change. If not, then please mention that the data are described in the Methods section. I was left wondering what lightning data you used.

Measuring FlashNet’s success

- I understand your motivation for using precision and recall as evaluation metrics. However, as a meteorologist, I lack an intuitive feeling for them, and I’m more used to looking at “probability of detection (POD)” and “false alarm ratio (FAR)”. Now, recall is the same as POD, so I’d recommend mentioning in the text that recall is also known as probability of detection or hit rate (these are terms used in the classic Wilks “Statistical Methods in the Atmospheric Sciences” book). The false alarm ratio is defined as $FP / (TP+FP)$, and I request that you please add this metric to Table 1. Note that false alarm rate, which is $FP / (FP+TN)$, is a poor choice for an imbalanced dataset such as this since most points are true negatives.

Assessing FlashNet's reliability

- I was left confused with the sharpness diagrams in Figure 4. To me, the insets looked like probability distribution functions (PDFs). I see the Met Office webpage says, "sharpness diagrams show the relative frequency with which the event has been predicted ... with different levels of probability". So, is that the same thing as a PDF, or is there some difference? Can you please provide a bit more information in the text so that readers can understand what sharpness diagrams are, and what they should look like – i.e., what constitutes a "good" sharpness diagram for this dataset?
- In Fig. 4, it does appear that one point for 2020 (red curve) falls outside of the skill area at low probability for 0-24h, and for 24-48h one point for 2020 falls outside at high probability. Was there something unique going on for your region in 2020? Your discussion in this section makes it sound as if the AI forecasts are always better, but I think you should mention these exceptions.

FlashNet against HRES

- "To the best of our knowledge; no previous attempts..." This is surprising, but I think is correct if you only consider peer reviewed literature. Daehyun Kim presented results on this at the 2021 AGU Meeting: "A14G-03 - A dynamical forecast-machine learning hybrid system for lightning prediction". Phillip Bothwell presented results at the 2010 International Lightning Detection Conference doing lightning prediction from NPW fields using a (non-ML) statistical method "EVOLUTION OF THE EXPERIMENTAL/AUTOMATED PERFECT PROG LIGHTNING FORECASTS AT THE STORM PREDICTION CENTER".

Discussion

- "we have baptized FlashNet" The word "baptized" sounds awkward in English, I would recommend, changing to "called", "dubbed", or "named" or something else similar. Even "christened" would be better, but still captures your connotation.
- "very useful for decision making" – you use this phrase in quotation marks twice in the paper, which makes it seem as though you are capturing the feedback from a human forecaster. Is that the case, and if so, please elaborate? If not, I would suggest dropping the quotation marks.

Methods

- I don't see anywhere that you mention what loss function was used in training your neural network.
- The ensemble-based AI approach used is novel and interesting, and I recommend mentioning it in the main text, because it is easy to overlook in the Methods section.

Answers to Reviewer #1 comments

We want to express our sincere gratitude for your thorough review of our manuscript. Your feedback has been instrumental in improving the quality and clarity of our work.

We are pleased to inform you that we have addressed all your concerns point by point. Your expertise and attention to detail have greatly contributed to strengthening our manuscript.

Thank you for your dedication to maintaining the quality of scientific publications.

Changes in the revised manuscript are reported in red color.

1) Reviewer

1. This paper shows the prediction capability of the lightning occurrence based on the FlashNet, and provides evaluation on the model performance regarding to different horizons. The result presentation seems sound. However, a more interesting aspect for the researchers in the field would be, what scientific factor has assured the model performance in 24-48 h horizon in the scientific view (for example, Figure 3-5, Table 2). For example, what is happening in the predicting horizon on 9 and 33 hours, when both recall and PRC-AUC reach peak? Please try to tell the stories behind figures and share the significance of science.

1) Authors

Thank you for the suggestion. We had noticed the diurnal cycle patterns in the error indices shown in Figures 3 and 5. These patterns are common to both the HRES forecasts and the FlashNet predictions. We had chosen not to comment on this behavior, as we considered it a characteristic inherited from HRES rather than a genuine effect of our strategy. The peaks highlighted above indeed reveal poorer skills of HRES during nighttime hours compared to daytime hours. This phenomenon is well-documented in the literature (Savazzi et al., 2022) and is attributed to insufficient nighttime convection, a known major shortcoming in IFS cycle 47r2 forecasts of convective activity. Since the skills of our hybrid strategy are correlated with those of HRES (from which FlashNet utilizes features), the observed peaks in the error indices in Figures 3 and 5 regarding our architecture are not surprising. Nevertheless, we agree on the importance of discussing this aspect in the revised version of our manuscript, and we have done so accordingly.

2) Reviewer

2. It seems to me that the outstanding of the model performance is largely dependent on the balanced dataset (from Figure 2). However, the dependence on the balanced dataset would be highly variable in terms of the machine learning tools. For example, in Mostajabi et al., which is a tree-based machine learning tool, model shows comparable performance to the PRC curves in Figure 2 as their unbalanced dataset are only slightly pre-processed. In that case, the main technical contribution of this research would be the optimization of the applicability on the neural networks to predict lightning rather than the integration of AI and NWP. Please provide more discussions on this issue.

Reference:

[1] Mostajabi, A., Finney, D.L., Rubinstein, M. and Rachidi, F., 2019. Nowcasting lightning occurrence from commonly available meteorological parameters using machine learning techniques. *Npj Climate and Atmospheric Science*, 2(1), p.41.

2) Authors

The analysis of this point is quite delicate. Contrary to what may be inferred from Figure 2, it is not at all true that our model's performance is strongly dependent on the dataset's imbalance. In fact, the opposite is true: our predictive model performs comparably whether it is applied to a balanced or an imbalanced dataset. This is a crucial aspect that perhaps should have been emphasized more in the manuscript. We did that in the revised version of our manuscript. We are very grateful to the reviewer for bringing this to our attention.

The crux of the matter is discussed in our section 'Explicit Formula for the Dependence of Precision Score on Prevalence,' particularly in Eq. (7). This formula, which does not appear to be documented in the literature, explains how the Precision-Recall (P-R) curve scales when transitioning from one prevalence value (and therefore, a specific level of imbalance) to another **while assuming the model's predictive abilities invariant by changing the prevalence**. Note that one does not need to rescale the Recall values, these latter being invariant under variations of prevalence.

The low AUC values in the case of the imbalanced dataset are thus simply a consequence of the high dataset imbalance, **not an indication of the poor predictive capabilities of our model**. To demonstrate the validity of this statement, in the revised manuscript we used Eq. (7) to rescale the P-R curves from the case with very low prevalence (all around 1%, curves and symbols in red in Fig. 2) to the corresponding balanced cases with a prevalence around 50%. The rescaled curves are depicted as orange dot-dashed and dashed lines and perfectly align with the blue lines. This serves as evidence for the truthfulness of our earlier assertion.

In the reference suggested by the Reviewer we do not see a similar evidence of model skill invariance by changing the dataset prevalence.

If we were provided with the AUC curve obtained by Mostajabi et al (2019), it would be straightforward for us to make a prevalence-adjusted comparison using the equation (7) from our study.

3) Reviewer

3. The presentation of this paper is mostly good, while the section “more on previous works: two different competing paradigms of prediction” is confusing. Since the introduction part has already compared the previous studies, including AI, NWP and AI integrated with NWP, the organization would cause redundancy to the description. It would be better to highlight different aspects of the research and provides the insight into the novelty.

3) Authors

Thank you for the suggestion: to make the manuscript clearer we merged the Introduction with the section titled “More on previous works: two different competing paradigms of prediction”.

4) Reviewer

4. Considering the applicability of the model, it would be better if more evaluations under different scenarios are provided. In the temporal aspect, most other studies would focus on certain months of the year when lightning occurrence tends to be active. For example, Sarkar et al. and Song et al. focused on MAM and JJA respectively. It would be of high interest if the model developed in this paper can perform well through the year and achieve low FAR even when lightning occurrence is very rare (for example, winter, early spring). Also, how well the model can nowcast the lightning occurrence when lightning has not been observed for long consecutive hours? With respect to the prediction results that are unreliable, what are the features that causes the wrong prediction result?

4) Authors

Thank you for the suggestion on focusing on certain months other than providing a cumulative analysis where all months have been considered all together as we did in our manuscript. According to the Reviewer's suggestion we have decided to show the Recall score reported in Fig. 5 for both HRES and FlashNet by fusing all folds and forecast lead times and then accumulating FP, TP, TN and FN for each month of the folds. In doing that, the resulting indices are shown month by month and separately for sea and land regions. Similar curves have been added for Precision=1-FAR. The lower skill score associated with Precision should not be a cause for concern, as it is a consequence of the high imbalance in the dataset as we have already discussed in our answers 2). What is crucial to note is that the skill score of FlashNet's Precision is always positive. A new figure (Fig. 6 in the revised manuscript) has been added to host these curves together with dedicated comments.

As for the last two raised issues, it should be noted that the imbalance in our dataset results from two types of imbalances: one related to the granularity in spatial lightning activity and the other pertaining to time granularity, a property mentioned in the Reviewer's report.

Focusing on the latter, we analyzed for this report Prevalence based on time series of randomly space-sampled points, and found it is centered around a prevalence of approximately 1%. This approach aligns with the spirit of the Reviewer's suggestion: in the majority of cases (around 99%), there is no observed lightning occurrence for extended periods (in line with the Reviewer's phrase 'lightning has not been observed for long consecutive hours'). The indices we have selected serve to assess the skills of our strategy under these conditions. What we did in the manuscript, thus perfectly fits the Reviewer's concerns related to the issue of forecasting lightning occurrence under a severe temporal-granularity of lightning occurrence.

To conclude on the issue of 'what are the features that cause incorrect prediction results?' it would be very interesting to analyze which features are responsible for errors in predictions. Unfortunately, conducting such an analysis in our case is not feasible due to the enormous number of features used, totaling around 500. Typically, such analyses, which are certainly very useful for a thorough understanding of which features have a fundamental impact on the results, are feasible in predictive models that use a limited number of features, as seen in the very interesting work of Mostajabi et al. (2019).

Minor Comments:

1. I'm not sure if it is the formatting issue but the figure next to "this is the resulting reliability diagram" is not seen in the document.

Authors: We rephrased as: "These are the instructions we followed to construct the reliability diagram."

2. Before the result section, the advantages of the proposed model are impressive. However, this paper seems only rationalized the undersampling issue, while the prediction with extreme meteorological events and the unseen encounter have not been explained in the subsequent sections.

Authors: Actually, the occurrence of lightning events refers, per se, to extreme (rare) events: lightning occurs indeed only in less than 1% of cases. The fact that FlashNet overcomes HRES both in minimizing the outcomes where the model incorrectly predicts the event occurrence (i.e. the False Positive events) and the outcomes where the model incorrectly predicts the negative class corresponding to the absence of the event (i.e. the False Negative events) is a clear indication of FlashNet's superiority in relation to prediction of unseen extreme events.

3. The explanation of the inset of Figure 4 seems confusing. Please rephrase the sentence.

Authors: We have rephrased the caption of Fig. 4 to clarify as suggested.

4. The sentence " LAMPINET network can reach a detection efficiency of 90% for intensity (I) > 50 kA, and location accuracy of 500 meters over the whole Italian territory, thus much higher than the HRES' spatial resolution." also needs citation

Authors:

We added the citation as suggested.

5. The parameters (for example, the number of hidden layers) of the ensemble model, as shown in Figure 6(b), should be provided.

Authors: We added this information as suggested in the new dedicated section “Neural network details”.

6. “The skills of our network are thus an indirect way to assess the skills of HRES in relation to meteorological variables triggering the lightning occurrence.” The rationale is needed.

Authors: We provided the rationale as suggested.

Answers to Reviewer #2 comments

We want to express our sincere gratitude for your thorough review of our manuscript. Your feedback has been instrumental in improving the quality and clarity of our work.

We are pleased to inform you that we have addressed all your concerns point by point. Your expertise and attention to detail have greatly contributed to strengthening our manuscript.

Thank you for your dedication to maintaining the quality of scientific publications.

Changes in the revised manuscript are reported in blue color.

1) Reviewer

It is a very interesting, timely and important study. The results are convincing and well presented. The methodology used is also well described.

The paper deserves to be published.

1) Authors

Thank you very much for your very positive feedback. We greatly appreciate your assessment of our study. Your encouragement is invaluable to us, and we are delighted to hear that you find our work both interesting and timely.

2) Reviewer

1. While authors have used 3 years of data set for the whole year and compared the forecast by HRES and Flashnet, it is worth to show the complete frequency distribution or probability distribution of lightning flashes from observation and from forecast (deterministic and AI)

2) Authors

It should be noted that our work aims to predict the probability of lightning occurrence, specifically, the presence of at least one lightning event (denoted with label 1). This focus differs from predicting the probability distribution of lightning abundance, as discussed explicitly in the paper's Discussion section. Consequently, it is logical for us to calculate $P(1)$ for HRES, our AI strategy, and observations. $P(1)$ represents the number of events with at least one lightning occurrence, normalized to the total number of events. This quantity is commonly referred to as Prevalence. Rather than reporting the actual Prevalence and its reconstruction from HRES and FlashNet predictions separately, we present these differences in the form of the BIAS index in Tab. 3 (also see our response to question 4).

3) Reviewer

2. Authors may provide the spatial distribution of lightning flashes as observed and predicted by the AI and HRES

3) Authors

Thank you for the suggestion, and, as per your recommendation, we have fused the three test folds and aggregated all the resulting 0s and 1s for each trimester of the resulting 'typical' year. We then displayed in a new figure (Fig. 7 in the revised manuscript) the accumulated values across the entire analyzed domain. This aggregation step is necessary due to the rarity of lightning occurrence, which account for less than 1% of the total events. For each trimester, we obtained three maps (one for HRES, one for FlashNet, and one for observations) that we presented together in Fig. 7, facilitating visual comparison. This holistic view appears to qualitatively affirm what all the quantitative statistical metrics we employed had previously established: the superiority of FlashNet over HRES in predicting lightning occurrences.

4) Reviewer

3. It is also worth to mention whether there is any systematic bias in the observed data sets etc.

4) Authors

We have added in the revised manuscript a new table (Tab. 3) reporting the BIAS of both HRES and FlashNet against the observations. See our answer 2) for the definition of BIAS. In this case as well, a clear superiority of FlashNet over HRES emerges. We have appreciated your suggestion.

5) Reviewer

4. As the region chosen has coastal area as well as mountainous region, it will be worth mentioning whether the HRES and AI both equally perform well in these heterogeneous spatial region or there is some improvement by one over another

5) Authors

Thank you very much for your suggestion; we greatly appreciate it. It is much easier for us to identify sea and land regions, and evaluate the model's performance for them separately. We hope this addresses the question raised by the Reviewer. There are indeed situations where coasts and mountains can be hardly identified or separated. The best example is the Liguria region, where one transitions from the sea to mountains reaching 2000 meters above sea level in less than 10-15 kilometers (i.e. the HRES grid-box). This information is provided in a new dedicated figure (Fig. 6 in the revised manuscript).

Answers to Reviewer #3 comments

We want to express our sincere gratitude for your thorough review of our manuscript. Your feedback has been instrumental in improving the quality and clarity of our work.

We are pleased to inform you that we have addressed all your concerns point by point. Your expertise and attention to detail have greatly contributed to strengthening our manuscript.

Thank you for your dedication to maintaining the quality of scientific publications.

Changes in the revised manuscript are reported in violet.

1) Reviewer

This paper presents an hybrid (physics-based combined with AI-based) method for lightning prediction in medium-range, e.g. 2 days. The method is simple and contains two steps: (1) using the state-of-the-art physics-based model (i.e. the ECMWF-HRES model) for extracting meteorological features, (2) feeding these features into a trained neural network for final prediction. According to the experiments, the prediction accuracy is higher than only using the ECMWF-HRES model.

Overall, this paper is interesting and provides useful results to the real-world applications. I evaluate the paper in the following aspects.

1) Authors

Thank you very much for your very positive feedback. We greatly appreciate your assessment of our study. Your encouragement is invaluable to us, and we are delighted to hear that you find our work both interesting and provides useful results to the real-world applications

2) Reviewer

1. This paper shows the insight that physics-based method and AI-based method shall be combined towards more accurate prediction. This is a promising direction provided that AI-based methods for weather forecasting have emerged in the past few years.

2) Authors

We appreciate Referee's acknowledge of the potential benefits of integrating both traditional meteorological knowledge and the power of AI in order to improve the accuracy of weather predictions.

3) Reviewer

(1) I am not sure if the authors tried to directly use the features extracted by the AI-based method as the input for the neural network. In particular, an open-sourced model, Pangu-Weather (which the authors know about and cited in the manuscript), has surpassed the ECMWF-HRES in the reanalysis data and shown good results in IFS initial data. I wonder if this implies better meteorological features can be extracted and used for lightning prediction (as well as other applications). Although AI-based methods mostly provided features in a lower-resolution, they can be simply interpolated (or not, but using neural networks for processing) for the purpose in this paper. I urge that such experiments are performed and compared to the proposed method. By the way, I understand that these AI-based models do not hinder the contribution of the proposed method, but the contribution needs to be further justified by the mentioned experiments.

3) Authors

We utilized authentic forecast data from HRES and employed them as input features for FlashNet. Our primary objective, as explicitly stated in the paper's title, is to demonstrate that the combination of deterministic models and AI methods synergistically yields a substantial improvement in

predicting extreme events at a low computational cost. Lightning events are, indeed, considered extreme events, as their occurrence is limited to less than 1% of the total.

The Referee's idea of utilizing Pangu-Weather, a revolutionary algorithm in the way meteorological forecasts are made, is truly exciting. If indeed Pangu-Weather exhibits superior skills compared to HRES, it is reasonable to anticipate further improvement if the features of Pangu-Weather were used as inputs for our network. However, we encounter several practical issues in carrying out such an exercise within this study. The first is that our network utilizes approximately 500 features, and these features should be present in their entirety as the output of Pangu-Weather. Using a subset of them could lead to different performances, and at that point, how would we attribute the cause? Is it caused by less effective learning when using Pangu-Weather fields, or is it due to the differing number of features? In short, it is a complex and delicate step that undoubtedly should be addressed in the future. We have added a comment in the discussion section regarding this aspect. We are fully open to collaborating with the inventors of Pangu-Weather to advance this study in the suggested direction.

4) Reviewer

(2) The neural network architecture seems not well discussed. There are many design choices, such as the number of layers, the choice of convolution against transformers, the dimension (number of neurons) of hidden layers, and many more. Without sufficient discussion based on ablation and/or diagnosis, I am not sure about the superiority of the presented design.

4) Authors

Thank you for the suggestion. Our approach aimed to minimize technical details in the text while ensuring the reproducibility of the network by making it available at the Zenodo link. However, we can attempt to strike a balance between an excess of technical details and easy access to fundamental information. We have added these technical details in the new dedicated section "Neural network details".

However, we believe it is crucial to clarify a key point regarding the main motivation of our work. Nowhere did we claim that our AI network is superior to other AI networks used in lightning nowcasting. Our objective was not to obtain the best possible AI network. On the contrary, our goal was to demonstrate that with a relatively simple AI network and numerous (accurate) features from a world-leading deterministic model in global-scale meteorological forecasting, significantly better results for lightning occurrence (and perhaps more generally for extreme event prediction) can be achieved **compared to what is directly produced by the deterministic model**. We believe that this objective has been widely achieved. We are confident that our work will inspire AI professionals to implement even more performant AI networks, thereby raising the bar for the goals we set in our seminal work.

5) Reviewer

3. The experiments also need some further justification.

(1) The data were collected in a limited region of the Earth and contains a small number of lightning events. I am not sure if the dataset is very easy for AI to fit and learn. Additionally, this is a highly imbalanced dataset, which is natural since lightning is kind of a rarely happening event. I am wondering if AI just learns to amend kind of bias from physics-based models (the ECMWF models were not tuned to fit lightning but aimed to offer global weather forecasting). If my hypothesis is true, this will not hinder the contribution of this work, but the paper needs further investigation and explanation on this topic.

5) Authors

The concepts of 'limited' and 'small number' are clearly relative. The area in question covers approximately **1 million square kilometers**, and the number of lightning events during the training period is very close to **1 million**. Therefore, these are not small numbers, especially considering that

the training was conducted with observed data as a target. We are not aware of similar studies on lightning that cover such an extensive area, with such a large number of lightning events, and over such an extended analysis period. However, there is no doubt that the dataset is extremely imbalanced. For this reason, we have developed an undersampling technique aimed at enabling the network to learn both the absence and presence of lightning events without introducing a potential bias towards class 0 (no lightning) during the learning phase, due to the significant dataset imbalance.

We would like to assure the Reviewer that our model not only eliminates the bias (an index that, in passing, was missing from the manuscript but has now been added to a new dedicated table, Table 3). The fact that this is not a simple bias removal process is evident from the fact that **FlashNet does not use lightning data from HRES among its features**. Therefore, our network does not involve a calibration process. Instead, the network learns to map other HRES features (convection proxies) to lightning events. We would struggle to define this type of process as anything other than a learning process.

6) Reviewer

(2) All experiments were performed in 0-24h and 24-48h, but both physics-based and AI-based methods have the ability of longer-range forecasting. It would be good to evaluate the method for lightning prediction in 3-5 days or even longer.

6) Authors

We fully agree with the Reviewer that technology exists to extend the forecasting horizon beyond 48 hours. However, we believe that the true value of our work lies in emphasizing what we have already highlighted in our response 4). Extending the forecast horizon to 3/5 days would be a brute-force approach to demonstrate something widely expected: that the skills of FlashNet will continue to outperform those of HRES, as clearly indicated by the transition from 0-24 to 24-48 hours (Fig. 5). Nevertheless, we would be very pleased if this work will trigger further investigations, including brute-force analyses, to move toward 10 day forecast or seasonal forecasts.

7) Reviewer

(3) I read the experiments carefully; it offered extensive results, but lacks analysis on why the proposed method can improve the results of ECMWF-HRES significantly. This may be closely related to (1) where a possible explanation lies in the amendment of bias, yet further analysis and discussions are needed.

7) Authors

As already emphasized in our response 5), FlashNet demonstrates its ability to learn how to map large-scale convection proxies (approximately 500 meteorological land/columnar observables) produced by the deterministic model HRES to the occurrence of at least one lightning strike on each node of a grid with a spacing of 10 km x 10 km over an area of approximately 1 million square kilometers. Why is this result successful? A significant portion of the credit goes to HRES: as the best existing deterministic model, it provides excellent large-scale proxies for lightning. We refer to thermodynamic variables, cloud microphysics, and convection descriptors. Another part of the success is attributed to our experience as Atmospheric Physicists in identifying 500 crucial features for lightning initiation from a much vaster set of possible features available in HRES output. Finally, despite its simplicity, the architecture has proven to be extremely robust against overfitting, exhibiting excellent generalization of its performance from one fold to another. Hence, there are various reasons behind the achievement of significant scientific interest.

Why does not the lightning field provided directly by HRES (variable litota3) have a similar quality to our prediction? Here too, the answer is simple: HRES uses a deterministic rather simple lightning initiation model that is, as result, unable to fully capture the complexity of the initiation phenomenon. A complexity, which the deep nonlinearity of our network has been able to capture

8) Reviewer

4. The paper seems not well organized and elaborated.

(1) The introduction (including the second part, "more on previous works...") seems too long. It would be better to squeeze this part and leave details to the appendix.

8) Authors

Thank you for the suggestion. We agree and we have reorganized the introduction accordingly.

9) Reviewer

(2) The main article contains little description about the proposed method, but simply lists experiments. I think basic descriptions about the proposed method shall be put into the main part.

9) Authors

Thank you for the suggestion. We agree and we have added a short summary of the proposed methodology before passing to discuss the results in detail.

10) Reviewer

(3) Important technical details and implementation details shall be put into the appendix. This includes but not limited to the description of network architecture, the length of training, how the loss function behaves, etc.

10) Authors

A new dedicated section, named "Neural network details", containing the technical details on our AI network has been added according to the Reviewer's suggestion.

11) Reviewer

(4) There are issues about the submitted code. The paper offers a link with DOI, but a single ZIP file (23GB) is provided, and it is somewhat difficult for me to download it.

11) Authors

We divided the original zip file into smaller zip files for easier downloading.

12) Reviewer

5. There are some typos in the paper. For example, in Page 3, "with different level*s* of maturity". Please check the manuscript carefully.

12) Authors

Thank you for the suggestion.

Answers to Reviewer #4 comments

We want to express our sincere gratitude for your thorough review of our manuscript. Your feedback has been instrumental in improving the quality and clarity of our work.

We are pleased to inform you that we have addressed all your concerns point by point. Your expertise and attention to detail have greatly contributed to strengthening our manuscript.

Thank you for your dedication to maintaining the quality of scientific publications.

Changes in the revised manuscript are reported in brown.

1) Reviewer

Overall

- This is a very good paper exploring the use of AI for predicting lightning at longer lead times. It is suitable for publication in Nature Communications after Minor Revision.

1) Authors

Thank you very much for your very positive feedback. We greatly appreciate your assessment of our study. Your encouragement is invaluable to us, and we are delighted to hear that you find our work suitable for publication in Nature Communications after Minor Revision.

2) Reviewer

- There are several instances (particularly in the “More on previous works”, “Results”, and “Measuring FlashNet’s” sections) where there are carriage returns, as if the end of the paragraph, but no tab to start the next paragraph. To me it seems that most or all of these are not intended to mark a new paragraph, but maybe a copy/paste or text encoding issue. Please fix.

2) Authors

Thank you for bringing the issue to our attention, which was related to a problem with LaTeX. We have fixed the issue wherever we found it.

3) Reviewer

Abstract

- “The prediction proficiency...” This is awkward in English – I would recommend replacing “proficiency” with “accuracy” or similar word.

3) Authors

Thank you for the suggestion. We followed your recommendation replacing proficiency with “capability”. Indeed “accuracy” is also a score index and could cause ambiguities.

4) Reviewer

Introduction

- “However, to set all free parameters, ... lightning flashes being rare events.” This sentence caused me consternation because, with 44 flashes occurring every second globally, it isn’t true that lightning flashes are rare events. I think your point is that lightning is rare in a machine learning sense, that is, a set of sample images over a region of interest will have pixels with lightning only rarely. So, I think what is missing from this sentence is that to set all free parameters “over a region of interest” ... This also gets at the concept that lightning relationships to meteorology may vary from region-to-region, which is further motivation for a machine learning based approach.

4) Authors

Thank you very much for correctly specifying the meaning of “rare event”. We have added what you suggested to the certainly incomplete sentence

5) Reviewer

- "...focus is restricted to the sole nowcasting" This is awkward in English; I would recommend changing to "focus is restricted solely to nowcasting".

5) Authors

Thank you for the suggestion. We followed your recommendation.

6) Reviewer

- "When NWP models are used in synergy with AI methods, it turns out that the forecast horizon was always limited to a few hours ahead." Why is this? The focus of the studies is on nowcasting, but I don't see any reason why NWP/AI synergy should be limited to nowcasting. This is confusing, please reword.

6) Authors

Thank you for having pointed out this issue. As we have reorganized the introduction following a recommendation from one of the four Reviewers, the highlighted sentence is no longer present in the revised manuscript.

7) Reviewer

More on previous works

- "Also, ERA5 can hardly been interpreted..." Grammar: change "been" to "be".

7) Authors

Sure. Thank you.

8) Reviewer

Results

- You mention the Methods section – can this be a "Data and Methods" section instead? If so, please make that change. If not, then please mention that the data are described in the Methods section. I was left wondering what lightning data you used.

8) Authors

We have used a standard wording in Nature where the Methods section essentially contains everything that is ancillary to obtaining the results. We suggest keeping the name of this section as 'Methods,' leaving the final decision on any name change to the editors. We would have no issues with changing it to 'Data and Methods,' as you have suggested.

9) Reviewer

Measuring FlashNet's success

- I understand your motivation for using precision and recall as evaluation metrics. However, as a meteorologist, I lack an intuitive feeling for them, and I'm more used to looking at "probability of detection (POD)" and "false alarm ratio (FAR)". Now, recall is the same as POD, so I'd recommend mentioning in the text that recall is also known as probability of detection or hit rate (these are terms used in the classic Wilks "Statistical Methods in the Atmospheric Sciences" book). The false alarm ratio is defined as $FP / (TP+FP)$, and I request that you please add this metric to Table 1. Note that false alarm rate, which is $FP / (FP+TN)$, is a poor choice for an imbalanced dataset such as this since most points are true negatives.

9) Authors

We completely understand your observation, but in fact, the metrics we have considered are exactly the ones you are referring to: $POD=Recall$, as you rightly pointed out. Additionally, we have $1-FAR = 1-[FP / (TP+FP)] = Precision$. The relationship between Precision and FAR is therefore straightforward. Our suggestion is to emphasize this aspect when defining Precision and Recall in the Methods section.

10) Reviewer

Assessing FlashNet's reliability

- I was left confused with the sharpness diagrams in Figure 4. To me, the insets looked like probability distribution functions (PDFs). I see the Met Office webpage says, "sharpness diagrams show the relative frequency with which the event has been predicted ... with different levels of probability". So, is that the same thing as a PDF, or is there some difference? Can you please provide a bit more information in the text so that readers can understand what sharpness diagrams are, and what they should look like – i.e., what constitutes a "good" sharpness diagram for this dataset?

10) Authors

The literature does not abound with 'official' definitions of 'sharpness diagrams.' We decided to include this characterization, inspired by the excellent MetOffice website referenced by the Reviewer: <https://www.metoffice.gov.uk/research/climate/seasonal-to-decadal/gpc-outlooks/user-guide/interpret-reliability>, where it is defined as follow: 'The sharpness diagrams show the relative frequency with which the event has been predicted (over the reference period and at all grid points) with different levels of probability.' We used this exact same definition in our work, aiming to facilitate interpretation by meteorologists.

Regarding the comment on the shape of the sharpness diagrams we obtained, we explicitly stated (largely inspired by the MetOffice site) that 'Forecast systems that are capable of predicting events with probabilities different from the Prevalence frequency are said to have sharpness - and our forecasts thus exhibit sharpness. The same conclusion holds (not shown) for the other two years of the k-fold cross-validation.'

Why do our diagrams (as those shown in the MetOffice site) exhibit sharpness? Our diagrams are non-zero for abscissa values different from the value of the Prevalence, which is approximately 0.01, and for this reason, we claimed that 'our forecasts thus exhibit sharpness.'

We believe that all the necessary information was already contained in the manuscript. However, we have added the citation to the MetOffice website in the revised version of our manuscript to further facilitate the interpretation of the resulting diagrams.

11) Reviewer

- In Fig. 4, it does appear that one point for 2020 (red curve) falls outside of the skill area at low probability for 0-24h, and for 24-48h one point for 2020 falls outside at high probability. Was there something unique going on for your region in 2020? Your discussion in this section makes it sound as if the AI forecasts are always better, but I think you should mention these exceptions.

11) Authors

In the revised version of our manuscript, we have included confidence bars corresponding to the 95% bootstrap confidence interval around the mean for each point on the reliability diagram, as shown in Figure 4. The reliability diagram represents the mean observed frequencies within each of the 9 bins (as described in Methods) into which we divided the range of variability of the forecast probability. This additional information allows us to provide an indication of the variability of the mean. Upon analyzing the updated figure, it is evident that, for both points highlighted by the Reviewer relative to the year 2020, the confidence bar (barely) enters the skill area. This information has been incorporated into the revised version of our manuscript.

12) Reviewer

FlashNet against HRES

- “To the best of our knowledge; no previous attempts...” This is surprising, but I think is correct if you only consider peer reviewed literature. Daehyun Kim presented results on this at the 2021 AGU Meeting: “A14G-03 - A dynamical forecast-machine learning hybrid system for lightning prediction”. Phillip Bothwell presented results at the 2010 International Lightning Detection Conference doing lightning prediction from NPW fields using a (non-ML) statistical method “EVOLUTION OF THE EXPERIMENTAL/AUTOMATED PERFECT PROG LIGHTNING FORECASTS AT THE STORM PREDICTION CENTER”.

12) Authors

Thank you for bringing to our attention the two approaches found in conference proceedings. We now cite both of them even if the second one does not utilize AI. It is however useful in relation to the use of the reliability diagrams as a verification tool.

Unfortunately, in the first contribution, as it is an abstract, there is not enough information available to assess the soundness of the approach or to use its results for comparison purposes.

13) Reviewer

Discussion

- “we have baptized FlashNet” The word “baptized” sounds awkward in English, I would recommend, changing to “called”, “dubbed”, or “named” or something else similar. Even “christened” would be better, but still captures your connotation.

13) Authors

As Italians, our linguistic sensitivity is necessarily limited. We thank you very much for the assistance to avoid being clumsy. We have replaced baptized by dubbed.

14) Reviewer

- “very useful for decision making” – you use this phrase in quotation marks twice in the paper, which makes it seem as though you are capturing the feedback from a human forecaster. Is that the case, and if so, please elaborate? If not, I would suggest dropping the quotation marks.

14) Authors

We use this phrase in quotation marks twice in the paper because this definition is provided in Weisheimer, A. & Palmer, T. N. On the reliability of seasonal climate forecasts. J. Royal Soc. Interface 11, 20131162 (2014). In that paper, Authors classified forecast skill into five categories based on the slope of the reliability line in the reliability diagram. Our reliability diagrams fall in the category defined by the Authors as “very useful for decision making”.

15) Reviewer

Methods

- I don’t see anywhere that you mention what loss function was used in training your neural network.
- The ensemble-based AI approach used is novel and interesting, and I recommend mentioning it in the main text, because it is easy to overlook in the Methods section.

15) Authors

In the revised version of our paper, we have provided more details regarding the AI network we have developed and used. A new section (“Neural network details”) has been added for this purpose. These details could have been retrieved directly from the Zenodo link we provided. However, we agree that

including such information within the manuscript facilitates the reading and understanding of the strategy employed.

Regarding the question of moving the description of our AI-based approach from the Methods section to the Main section, we have some reservations. The main reason for this hesitation is that in our work, the AI network is instrumental in achieving more accurate results when used in synergy with HRES. In other words, the network, while being an original and innovative development of this work, is primarily a means to a final object and not the main outcome of the present study. Instead, the main contribution of our work lies in demonstrating that by using deterministic methods and AI methods in synergy, a significant added value can be obtained compared to the sole use of the purely deterministic approach. Coherently, the title of our paper specifically poses a question about this aspect.

REVIEWER COMMENTS

Reviewer #1 (Remarks to the Author):

The revised version of this article has to some extent resolved the comments, and explained the confusing part in the original manuscript. The overall quality of the manuscript has increased. Still, the applicability of the model and the scientific behind the model are core concerns, while the response to the comment 1 is not satisfactory and more details are anticipated. Please see the comments below:

Major comment 1:

The response to the first comment is quite intriguing, as the author pointed that the decreased model performance can be attributed to the intuitive defect of HREs during nighttime. Does it mean that the forecast is made only at certain time of the day (instead of anytime during the day)? Please rationalize it since I expect that the forecast horizon would not be associated with the local time. It seems confusing and please correct me if I have misunderstood it.

Major comment 2:

More significantly, from other published paper related to lightning, it could be suggested that the lightning would tend to occur more frequently during the nighttime (for example, Figure 5(c) in Song et al.). Thus, I consider it more important to forecast lightning with high accuracy at nighttime as well. However, it seems that the model still maintained the associated defects of the HREs model. Also, Song et al. have indicated a better prediction skill during the nighttime (Figure 5(d)). A feasible enhancement on the model to rid of the HREs defects might be introducing the real observations (LAMPINET) into the AI training process as well. It would also be interesting to discuss if the data imbalance helps or worsens the model prediction during the nighttime, where more lightning-active cases are trained.

Reference:

Song, G., Li, S. and Xing, J., 2023. Lightning nowcasting with aerosol-informed machine learning and satellite-enriched dataset. *npj Climate and Atmospheric Science*, 6(1), p.126.

Reviewer #3 (Remarks to the Author):

I read the rebuttal and appreciate the authors' efforts to have addressed most of my concerns. I leave a few minor comments below; they are numbered in the same way as they appeared in the rebuttal.

3) The neural networks are widely reknowned to process a large number of features, not limited to approximately 500. I am still wondering if there is any method to use the AI-generated features for lightning prediction. Any experiments are appreciated and can be put into the appendix if they look preliminary.

6) I agree that extending prediction to a lead time of 3/5 days is not difficult, but I expect some quantitative results, e.g. based on the same framework and setting and simply changing the lead time.

7) Is the final sentence in the rebuttal incomplete or not deleted from a draft? Additionally, I hope that this point (the response to 7) can be discussed in the paper.

I also read other referees' comments and it seems that all referees tend to agree with the contribution of this work. After the above minor concerns are addressed, I would be happy to have this paper accepted.

Reviewer #4 (Remarks to the Author):

My review comments have been adequately addressed.

Answers to Reviewer #1 comments

We want to express our sincere gratitude for your thorough review of our revised manuscript. Your feedback has been instrumental in improving the quality and clarity of our work.

We are pleased to inform you that, also in this case, we have addressed all your concerns point by point. Your expertise and attention to detail have greatly contributed to strengthening our manuscript. Thank you for your dedication to maintaining the quality of scientific publications.

Changes in the revised manuscript are reported in red color.

1) Reviewer

The response to the first comment is quite intriguing, as the author pointed that the decreased model performance can be attributed to the intuitive defect of HREs during nighttime. Does it mean that the forecast is made only at certain time of the day (instead of anytime during the day)? Please rationally since I expect that the forecast horizon would not be associated with the local time. It seems confusing and please correct me if I have misunderstood it.

1) Authors

Thank you for asking. The ECMWF conducts four daily runs of the HRES model. These runs start from the so-called 'analysis' (i.e. the initial conditions) at the synoptic hours 00:00 UTC, 06:00 UTC, 12:00 UTC, and 18:00 UTC. Among them, the 00:00 UTC run stands out as the most crucial, serving as the initial forecast for the day. It plays a pivotal role in shaping early morning and daytime weather predictions, making it of paramount importance in the operational weather model's daily cycle. This is why we have chosen the 00:00 UTC model run as our final preference. According to our choice, the local time (expressed in UTC) in this work coincides with the lead time of the forecast. We have added this information in the revised manuscript.

2) Reviewer

More significantly, from other published paper related to lightning, it could be suggested that the lightning would tend to occur more frequently during the nighttime (for example, Figure 5(c) in Song et al.). Thus, I consider it more important to forecast lightning with high accuracy at nighttime as well. However, it seems that the model still maintained the associated defects of the HREs model. Also, Song et al. have indicated a better prediction skill during the nighttime (Figure 5(d)). A feasible enhancement on the model to rid of the HREs defects might be introducing the real observations (LAMPINET) into the AI training process as well. It would also be interesting to discuss if the data imbalance helps or worsens the model prediction during the nighttime, where more lightning-active cases are trained.

Reference:

Song, G., Li, S. and Xing, J., 2023. Lightning nowcasting with aerosol-informed machine learning and satellite-enriched dataset. *npj Climate and Atmospheric Science*, 6(1), p.126.

2) Authors

We greatly appreciate the Reviewer for bringing the work of Song, G., Li, S., and Xing, J., 2023, to our attention. However, it should be noted that this work falls within the extensive literature on lightning nowcasting models, a forecasting regime that we explicitly mentioned in our introduction is not the primary focus of our study. **Nevertheless, the Reviewer's suggestion offers a potential avenue for further extending our work by incorporating observed features**, particularly observations from the LAMPINET network, to enhance short-term forecasting. In our study, as explained in our previous response, a short-term forecast refers to the initial few forecast hours (e.g., 1 or 2 hours after the analysis at 00 UTC) of the night following the 00 UTC analysis. This suggests that incorporating observations might provide some benefit for the first (or at least a portion of it)

forecasted night. Unfortunately, we firmly exclude the same potential benefit for the night in the 24-48 hour forecast horizon, as such a lead time is too distant from the observation time to maintain a relevant correlation with it. In the revised version of our manuscript, we have added a dedicated comment on this aspect (please refer to the 'Discussions' section), also citing the suggested reference by Song et al., 2023 along with an attempt we made in the recent past to integrate observations into a medium-range forecast pipeline.

Answers to Reviewer #3 comments

We want to express our sincere gratitude for your thorough review of our manuscript. Your feedback has been instrumental in improving the quality and clarity of our revised version.

We are pleased to inform you that we have addressed all your concerns point by point. Your expertise and attention to detail have greatly contributed to strengthening our manuscript.

Thank you for your dedication to maintaining the quality of scientific publications.

Changes in the revised manuscript are reported in violet.

1) Reviewer

3) The neural networks are widely reknowned to process a large number of features, not limited to approximately 500. I am still wondering if there is any method to use the AI-generated features for lightning prediction. Any experiments are appreciated and can be put into the appendix if they look preliminary.

1) Authors

We have already addressed this issue in the Discussion section regarding Pangu-Weather. As the Reviewer is certainly aware, adding new features to an existing neural network is generally not a highly challenging task. On the contrary, it is a labor-intensive process to gather data, adapt formats, conduct new training, and perform new testing, especially with the scientific rigor we have followed in the present work. Essentially, it constitutes a new project, undoubtedly interesting but one that we do not consider suitable for inclusion as a mere appendix. How to make it preliminary? **If a study is conducted following the path of scientific rigor, in our opinion, it cannot be considered preliminary to avoid conveying incomplete or, perhaps, misleading messages.** We regret that we cannot satisfy the Reviewer's curiosity, which, nevertheless, is shared by us. If the authors of Pangu-Weather provide us with features ready for assimilation into our network, it would be our pleasure to establish a collaboration and explore new research avenues.

2) Reviewer

6) I agree that extending prediction to a lead time of 3/5 days is not difficult, but I expect some quantitative results, e.g. based on the same framework and setting and simply changing the lead time.

2) Authors

Our answer is similar to the previous one. Indeed, we agree that extending the forecast horizon to 3/5 days is a conceptually straightforward exercise. However, from an operational standpoint, it demands a considerable effort. The data retrieval process for HRES model outputs alone is labor-intensive, and download speeds are an uncontrollable variable. At this stage of the review process, is it worth delaying the dissemination of our results to the scientific community by several months to add an outcome that, while potentially interesting, we consider secondary to the primary aim of our work? In our opinion, the cost-benefit ratio associated with the posed question is strongly skewed in favor of the former. It should also be emphasized that even within the 0-24 hour forecast interval, the literature does not offer AI-enhanced forecasting solutions. We have not merely filled this gap but have pushed beyond the 24-hour horizon. Therefore, we are already well beyond the existing state of the art.

3) Reviewer

Is the final sentence in the rebuttal incomplete or not deleted from a draft? Additionally, I hope that this point (the response to 7) can be discussed in the paper.

3) Authors

The final sentence in the rebuttal (point 7) was actually complete. We only missed the final 'dot' in the sentence. Thank you for the suggestion regarding adding the discussion raised in the previous rebuttal to the paper. It has now been included in the revised version (Discussions).

REVIEWERS' COMMENTS

Reviewer #1 (Remarks to the Author):

The authors have responded to my concerns with careful clarifications. The quality and the scientific contents of this paper meet the requirement of the journal. I have no further comments.

Reviewer #3 (Remarks to the Author):

I read the rebuttal. The third question has been well addressed.

Regarding the authors' response to the former questions, I understand that there are difficulties in performing the mentioned study, but I do not totally agree that the additional experiments can only add marginal values to the submission. I am a researcher in the AI field. When I review the paper from my viewpoint, I care about the generalized ability (lying at the core of AI) of the proposed method, reflecting how the method is scaled up (e.g. to use a larger set of features) or generalized (to a longer lead time).

If the first experiment (using more complex models based on AI-generated features) needs some inputs like assimilation, I understand and please discuss a bit in the paper (probably the authors have done so). I do not really understand why it requires heavy efforts (e.g. for a few months) to perform the second experiment (i.e. working on a longer lead time). If this is the case (months are needed), I will not choose to hold on the review process, considering that the main contribution is not impacted by missing the extension.

In summary, I do not have additional comments in this round. I do not resist accepting this submission in the current form, but I think that more experiments can make it stronger. It is up to the editor to decide.

Answers to Reviewer #3 comments

We want to express our sincere gratitude for your thorough review of our manuscript. Your feedback has been instrumental in improving the quality and clarity of our revised version.

Your expertise and attention to detail have greatly contributed to strengthening our manuscript. Thank you for your dedication to maintaining the quality of scientific publications.

Changes in the revised manuscript are reported in violet.

1) Reviewer

Regarding the authors' response to the former questions, I understand that there are difficulties in performing the mentioned study, but I do not totally agree that the additional experiments can only add marginal values to the submission. I am a researcher in the AI field. When I review the paper from my viewpoint, I care about the generalized ability (lying at the core of AI) of the proposed method, reflecting how the method is scaled up (e.g. to use a larger set of features) or generalized (to a longer lead time).

If the first experiment (using more complex models based on AI-generated features) needs some inputs like assimilation, I understand and please discuss a bit in the paper (probably the authors have done so). I do not really understand why it requires heavy efforts (e.g. for a few months) to perform the second experiment (i.e. working on a longer lead time). If this is the case (months are needed), I will not choose to hold on the review process, considering that the main contribution is not impacted by missing the extension.

In summary, I do not have additional comments in this round. I do not resist accepting this submission in the current form, but I think that more experiments can make it stronger. It is up to the editor to decide.

1) Authors

We fully understand the constructive spirit of your comments, including the one about studying the extension of our forecast to 72 hours. The problem we mention in relation to the heaviness of this further generalization is mainly related to the time required to download the HRES features across the entire territory under study and for the 3 years corresponding to the three folds on which the analysis was done. If it were just about downloading the litota variable, this would not be problematic. However, since it involves all the features listed in Table 4, the download times are significantly longer, making the activity lengthy. As a compromise, we have explicitly mentioned in the 'Discussions' section that this is nevertheless an interesting topic for future research.